# Colour polymorphism associated with a gene duplication in male wood tiger moths

**Melanie N Brien[1]\*[†], Anna Orteu[2][†], Eugenie C Yen[2], Juan A Galarza[3], Jimi Kirvesoja[4], Hannu Pakkanen[5], Kazumasa Wakamatsu[6], Chris D Jiggins[2][‡], Johanna Mappes[1,4][‡]**

[1]Organismal and Evolutionary Biology Research Program, Faculty of Biological and Environmental Sciences, University of Helsinki, Helsinki, Finland; [2]Department of Zoology, University of Cambridge, Cambridge, United Kingdom; [3]Ecology and Genetics Research Unit, University of Oulu, Oulu, Finland; [4]Department of Biological and Environmental Science, University of Jyväskylä, Jyväskylä, Finland; [5]Department of Chemistry, University of Jyväskylä, Jyväskylä, Finland; [6]Institute for Melanin Chemistry, Fujita Health University, Toyoake, Japan

**\*For correspondence:**
mnbrien1@gmail.com

[†]These authors contributed equally to this work
[‡]These authors also contributed equally to this work

**Competing interest:** The authors declare that no competing interests exist.

**Abstract** Colour is often used as an aposematic warning signal, with predator learning expected to lead to a single colour pattern within a population. However, there are many puzzling cases where aposematic signals are also polymorphic. The wood tiger moth, *Arctia plantaginis*, displays bright hindwing colours associated with unpalatability, and males have discrete colour morphs which vary in frequency between localities. In Finland, both white and yellow morphs can be found, and these colour morphs also differ in behavioural and life-history traits. Here, we show that male colour is linked to an extra copy of a *yellow* family gene that is only present in the white morphs. This white-specific duplication, which we name *valkea,* is highly upregulated during wing development. CRISPR targeting *valkea* resulted in editing of both *valkea* and its paralog, *yellow-e,* and led to the production of yellow wings. We also characterise the pigments responsible for yellow, white, and black colouration, showing that yellow is partly produced by pheomelanins, while black is dopamine-derived eumelanin. Our results add to a growing number of studies on the genetic architecture of complex and seemingly paradoxical polymorphisms, and the role of gene duplications and structural variation in adaptive evolution.

## Editor's evaluation

Through genetic mapping and analysis of WGS data, the authors identify a gene duplication co-segregating with a color polymorphism in males of the aposematic tiger moth. They name the new gene valkea and investigate its expression and function in relation to wing pigmentation. Using CRISPR to disrupt valkea, they observe changes in wing color. However, because valkea was not the only gene edited, its causal role in the color polymorphism cannot be unambiguously established.

## Introduction

Colour polymorphisms, defined as the presence of multiple discrete colour phenotypes within a population (*Huxley, 1955*), provide an ideal trait to study natural and sexual selection. Colour phenotypes can have an effect on fitness in many contexts including camouflage, mimicry, and mating success. Colour is often associated with aposematism, where it acts as a signal, warning predators

**Figure 1.** The wood tiger moth, *Arctia plantaginis*. (**A**) Sampling locations and frequencies of yellow and white males in Finland, Scotland, and Estonia. (**B**) Males of the white and yellow colour morphs (credit: Samuel Waldron).

of unpalatability (*Cott, 1940*; *Cuthill et al., 2017*). In such cases, predator learning should favour the most common colour pattern, leading to positive frequency-dependent selection (*Endler, 1988*). Despite this, aposematic polymorphisms can be stable when selection is context-dependent (*Briolat et al., 2019*), especially where genetic correlations between colour phenotypes and other traits lead to complex fitness landscapes (reviewed by *McKinnon and Pierotti, 2010*).

A variety of genetic mechanisms can underpin these types of complex polymorphisms involving multiple associated traits (*Orteu and Jiggins, 2020*). In many cases, such complex polymorphisms are controlled by 'supergenes' in which divergent alleles at several linked genes are maintained in strong linkage disequilibrium by reduced recombination. The most common mechanism for locally reduced recombination are inversions, which range from single inversions involving a small number of genes, to multiple nested inversions covering large genomic regions (*Joron et al., 2011*; *Wang et al., 2013*; *Küpper et al., 2016*; *Funk et al., 2021*). Nonetheless, other mechanisms for reducing recombination, such as centromeres or large genomic deletions, may also play a role. An alternative mechanism is that a single regulatory gene controls variation via multiple downstream effects (*Thompson and Jiggins, 2014*). While there are fewer instances in which multiple phenotypes seem to be controlled by a single gene, one potential example is the common wall lizard, where colour genes have pleiotropic effects on behavioural and reproductive traits (*Andrade et al., 2019*). Multiple mutations within a single gene can also lead to variation in multiple traits (*Linnen et al., 2013*).

The wood tiger moth, *Arctia plantaginis*, has a complex polymorphism that has been well studied in an ecological context. Males show polymorphic aposematic hindwing colouration with discrete yellow, white, or red hindwing colour morphs found at varying frequencies in different geographic locations. In Finland, for example, both yellow and white morphs can be found, with white morphs varying in frequency from 40 to 75% (*Galarza et al., 2014*). In Estonia, white morphs make up 97% of the population, while yellows morphs form a completely monomorphic population in Scotland

(*Hegna et al., 2015*; *Figure 1*). Long-term breeding studies of these moths have shown that male hindwing colour is a Mendelian trait controlled by a single locus with two alleles (*Suomalainen, 1938*; *Nokelainen et al., 2022*). White alleles (W) are dominant over the yellow (y). These colour genotypes also covary with behavioural and life-history traits, contributing to the maintenance of this polymorphism. Yellow males are subject to lower levels of predation in the wild (*Nokelainen et al., 2012*; *Nokelainen et al., 2014*), while white males have a positive frequency-dependent mating advantage (*Gordon et al., 2015*). There are differences in chemical defences, and bird reactions to these defences, between the genotypes (*Rojas et al., 2017*; *Winters et al., 2021*). Yellow morphs show reduced flight activity compared to white males, although yellows may fly at more selective times, that is, at peak female calling periods (*Rojas et al., 2015*). Increased reproductive success in Wy genotype females points towards strong heterozygote advantage (*De Pasqual et al., 2022*). In summary, there is a trade-off between natural selection through predation and reproductive success, which contributes to the maintenance of this polymorphism (*Rönkä et al., 2020*).

Despite the large body of research on *A. plantaginis* colour morphs, the genetic basis of this polymorphism is unknown. Here, we explore male hindwing colour variation using linkage mapping, whole-genome data, and gene expression analyses, with wild populations and lab crosses, to identify the locus controlling the colour polymorphism. We use CRISPR/Cas9 gene knockouts to determine the function of the identified gene, and then characterise the pigments producing yellow, white, and black colouration on the wings of male *A. plantaginis*. Our findings aim to provide an example of the genetic architecture controlling a trait that is part of a complex polymorphism.

## Results

### A narrow genomic region is associated with hindwing colour

To investigate the genetic basis of male hindwing colouration in *A. plantaginis*, we carried out a quantitative trait locus (QTL) mapping analysis using crosses between heterozygous Wy males and homozygous yy females. We used RADseq data aligned to the yellow *A. plantaginis* reference genome from 172 male offspring (90 white and 82 yellow) from four families. The QTL analysis identified a single marker associated with male hindwing colour (*Figure 2A*). This marker was found on scaffold YY_tarseq_206_arrow at position 9,887,968 bp (95% confidence intervals 9,349,978–9,888,009 bp) and had a LOD score of 32.8 (p<0.001). The significant marker explains around 75% of the phenotypic variation and, with one exception, yellow individuals all had a homozygous yy genotype at this marker.

To further narrow down this region, we ran a genome-wide association study (GWAS) using whole-genome sequences of males from four populations: polymorphic Southern Finland (5 white, 5 yellow) and Central Finland (10 white, 10 yellow), Estonia (4 white), where males are mostly white, and monomorphic Scotland (12 yellow), where all males are yellow. This identified a region of associated SNPs also on scaffold 206 (*Figure 2B*). Two SNPs, 137 bp apart (at positions 9,885,384 and 9,885,521), were significant above a strict Bonferroni corrected threshold. A total of 162 SNPs were over the threshold of p<0.0001 and, of these, 155 are within a 99 kb region on scaffold 206 (9,833,387–9,932,264 bp). The top SNPs are within 2.5 kb from the top QTL marker, and the SNP at this marker has a p-value<0.0001.

The 538 Kb QTL interval contains 21 genes (*Supplementary file 1A*) which were annotated with reference to *Drosophila melanogaster*. Of these genes, four are part of the *yellow* gene family. The top two SNPs from the GWAS, and the top marker from the QTL, fall in a non-coding region upstream of the gene, *yellow-e*, and are also close to an additional *yellow* gene, *yellow-g*.

### Identifying structural variation in this region

The trio binning method used by *Yen et al., 2020* to assemble the *A. plantaginis* reference genome produced two reference sequences, one for a white allele and one for a yellow allele. We extracted the region containing the QTL interval from the yellow reference and aligned it against the white reference. The alignment showed a duplicated region approximately 117 kb long on scaffold 419 of the white reference from around 6,941,000–7,058,000 bp (*Figure 2—figure supplement 1*). The *yellow-e* gene and its flanking regions are within this sequence and are therefore duplicated in the white reference (*Figure 2C*). One copy of the gene (named jg1310 in the W annotation) has seven exons and is similar to the *yellow-e* gene in the yellow reference (99.7% identity in coding sequences). The second copy unique to the white scaffold (jg1308) has only the first five exons (81.8% identical to the gene in

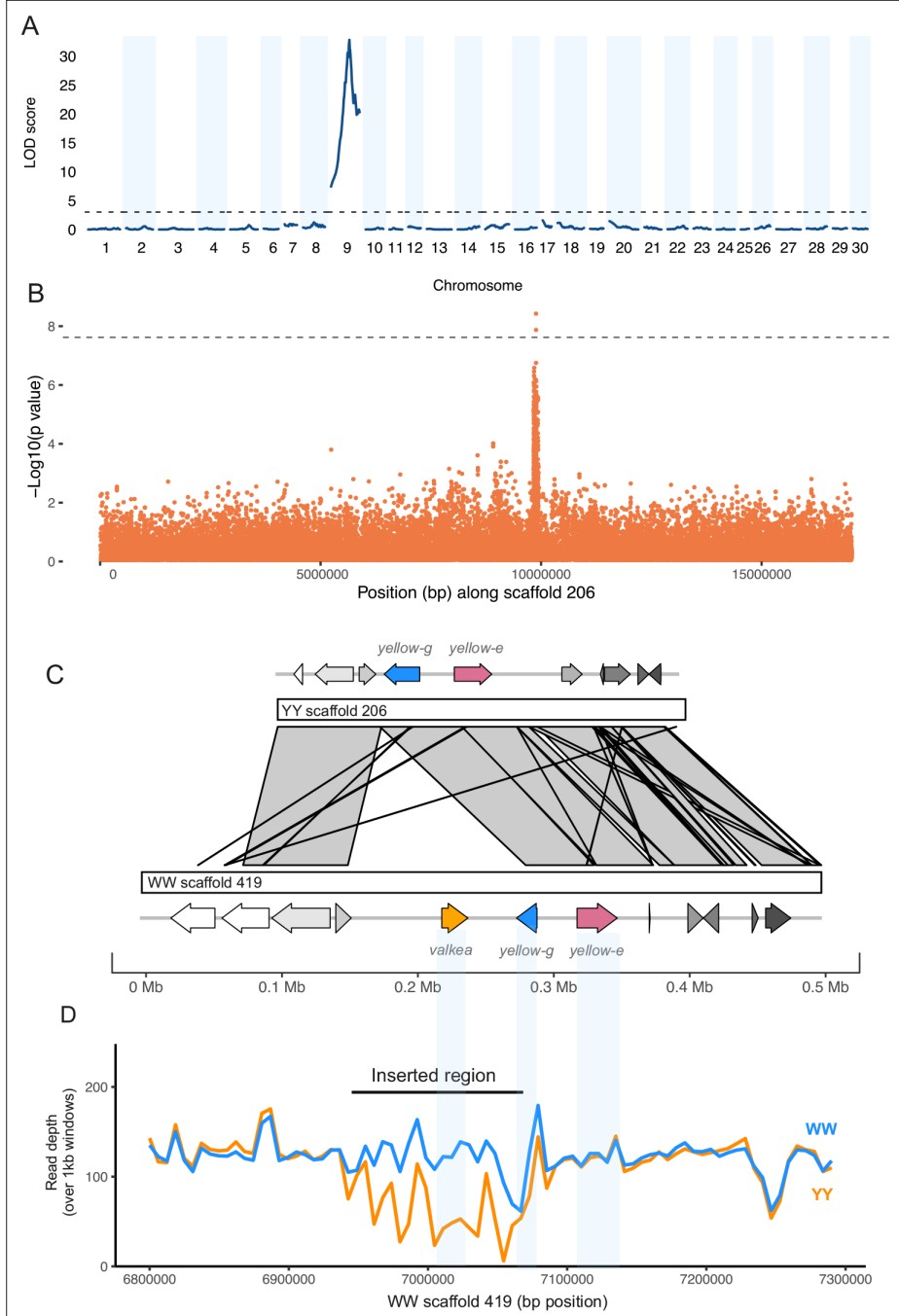

**Figure 2.** A duplicated region in white morphs is associated with male hindwing colour. (**A**) Quantitative trait locus (QTL) analysis of white and yellow F1 males (n = 172) reveals a 500 kb region significant on scaffold 206 of the yellow reference, part of linkage group 9. The dotted line indicates the significance threshold determined by permutation tests (p=0.05). (**B**) A genome-wide association study (GWAS) of wild samples (n = 46) showed SNPs associated with hindwing colour along the same scaffold. The dotted line shows the Bonferroni corrected significance threshold. (**C**) Alignment of the white and yellow reference genomes reveals an insertion in the white reference sequence that contains a copy of the *yellow-e* gene which we named *valkea*, in addition to the *yellow-e* present in both white and yellow morphs. (**D**) Mean read depth across the candidate region in all Finnish white (WW and Wy) and yellow (yy) samples.

The online version of this article includes the following source data and figure supplement(s) for figure 2:

**Figure supplement 1.** Dotplot of the alignment between a region of scaffold 419 from the white reference and scaffold 206 from the yellow reference.

*Figure 2 continued on next page*

*Figure 2 continued*

**Figure supplement 2.** Mapping quality (MQ) across the latter half of scaffold WW_tarseq_419_arrow.

**Figure supplement 3.** When filtering for increasing mapping quality, more reads are lost within the duplication in yellow samples compared to white samples, suggesting that reads in this region are more likely to be mismapped than in white samples.

**Figure supplement 4.** Example gel images for the genotyping primers.

**Figure supplement 4—source data 1.** Unedited gel image showing the genotyping primers.

**Figure supplement 5.** Tree of coding sequences for all *yellow* family genes found in *A. plantaginis* and *B. mori.*

the yellow reference), possibly due to a stop codon mutation in the fifth exon. For clarity, we named this duplicated white-specific copy *valkea*, in reference to a Finnish word for 'white'. While all white samples had consistent coverage of reads across the duplicated region, coverage was patchy in yellow samples, with many regions having no and very low coverage in yellow samples (*Figure 2D*). Those reads that map in the *valkea* region in yellow samples are likely to be mapping errors, because the sequence similarity is high and mapping quality is reduced within the duplication (*Figure 2—figure supplement 2*). When increasing the mapping quality filtering, read depth decreases more in yellow samples compared to white samples in this region (*Figure 2—figure supplement 3*). We confirmed the absence of this region in yellow individuals by designing primers within the duplication (*Supplementary file 1B*, *Figure 2—figure supplement 4*), which only amplified in WW and Wy samples, including Finnish, Estonian, and lab populations.

To confirm that both of these gene copies are related to *yellow-e*, we compared them to *yellow-e* orthologues found in *Bombyx mori*, *Heliconius melpomene,* and *D. melanogaster,* along with other *yellow* genes from *A. plantaginis* and *B. mori*. Both of the tiger moth genes were most closely related to the *H. melpomene yellow-e* (*Figure 2—figure supplement 5*). Between *valkea* and *yellow-e*, there is an additional gene which showed highest similarity to *Drosophila yellow-g2* (when extracted from both the white and yellow references). This gene is not part of the duplicated sequence and is present as a single copy in both morphs. Coverage across *yellow-g* and *yellow-e* genome regions in wild samples is similar in both morphs (*Figure 2D*). Upstream of the duplication is an unnamed gene (listed as jg6744 in the yellow annotation and jg1307 in the white). This is the same orthologous gene in both reference genomes, having 99.3% identity. Similarly, if we look at the 150 kb upstream region, sequence identity is 99.98%. There are no non-synonymous mutations between coding sequences of *yellow-e* when comparing the white and yellow references, although there are differences in the first exon of *yellow-g*. The absence of this duplicated region in the yellow morphs means we cannot determine if there is a change in linkage disequilibrium across white and yellow morphs.

### *Valkea* is differentially expressed between morphs

To pinpoint which of these candidate genes is associated with male wing polymorphism in *A. plantaginis*, we next performed gene expression analyses across several developmental stages. Based on knowledge of the expression patterns of *yellow* genes (*Ferguson et al., 2011*) and other melanin pathway genes such as *pale, ebony and ddc* in Lepidoptera (*Zhang et al., 2017b*), we hypothesised that changes in gene regulation that control the development of wing colour morphs in the tiger moth most likely occur during pupal development. Pupal development in the wood tiger moth lasts for approximately 8 days, and no colour is present in the wings until day 7, when the yellow pigment appears. A few hours later, black melanin pigmentation is deposited. We sampled two stages early in development when no colouration is present in the wings (72 hr post-pupation, and 5-day-old pupae), and two stages later in development: the point when yellow appears in yy morphs (Pre-mel, 7-day-old pupae) and the other after black melanin has also been deposited (Mel, 7–8-day-old). Forty individuals were sampled in total – five per genotype and stage.

First, we explored the general patterns of expression by mapping RNAseq reads to the white reference genome, which contains the duplicated region that includes *valkea*. We filtered out lowly expressed genes, retaining 10,920 genes and used multidimensional scaling (MDS), a dimensionality reduction technique, to explore which factors explain genome-wide variation in gene expression between samples. We observed that samples clustered based on their developmental stage, suggesting it is an important factor driving differences in genome-wide gene expression between

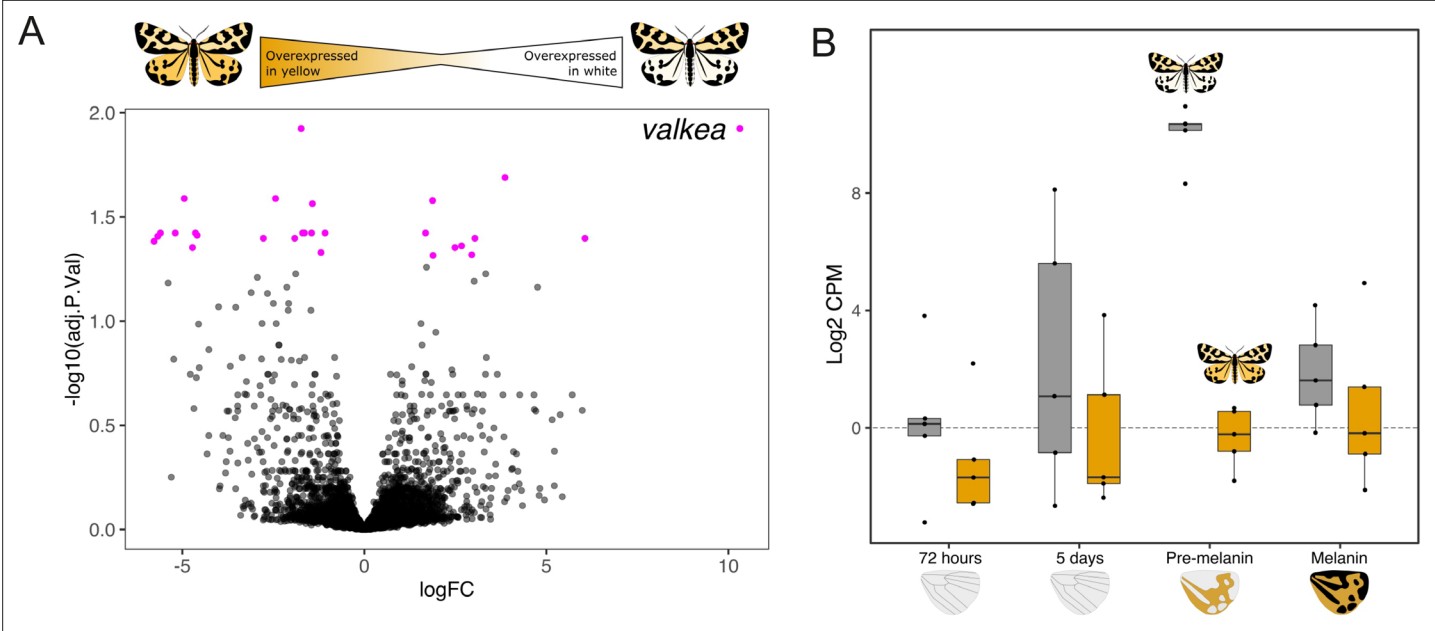

**Figure 3.** *Valkea* is overexpressed in white males in the pre-melanin stage. (**A**) In pink are genes that are significantly differentially expressed between yellow and white morphs at the pre-melanin stage. *Valkea* is the most differentially expressed (DE) gene (i.e. gene with the highest log fold change). (**B**) Expression of *valkea* across developmental timepoints shows that it has higher expression measured in Log2 CPM (counts per million) in white individuals compared to yellow ones. Expression of *valkea* in yellow morphs is around 0.

The online version of this article includes the following figure supplement(s) for figure 3:

**Figure supplement 1.** Genome-wide expression patterns are shaped by developmental stage.

**Figure supplement 2.** Expression of the gene 'jg15945' across the four developmental stages.

samples (*Figure 3—figure supplement 1*). Such a pattern would be expected as many genes are involved in development and thus are likely to be differentially expressed (DE) between developmental stages. No apparent clustering can be observed among samples of the same colour morph.

We next compared gene expression between yy and WW individuals at each of the developmental stages. Overall, 99 genes were differentially expressed (FDR < 0.05) between the two morphs (*Figure 3A*). Two of these DE genes, *yellow-e* and *valkea*, are two of the 22 genes identified in the GWAS and QTL analysis. *Valkea* was overexpressed in white individuals in the pre-melanin stage with a log fold change of 10.32 and a p-value of 2.18e-06. As *valkea* is only fully present in the W genome, it is not expected to be expressed at all in the Y genome. *Yellow-e* was also overexpressed in white individuals during the pre-melanin stage with a log fold change of 3.86 and adjusted p-value of 5.62e-06. In other developmental stages, neither *valkea* nor *yellow-e* showed differences in expression between morphs (*Figure 3B*).

Of the 99 genes differentially expressed between white and yellow individuals across development, 49 were upregulated in the yy morph, while the remaining 50 were upregulated in WW individuals. The earliest developmental stage, 72 hr, was the stage with the highest number of DE genes (n = 48), while the 5-day-old stage had the fewest (n = 7). One gene which encodes a C2H2 zinc finger transcription factor in *D. melanogaster*, 'jg15945', was over-expressed in yy in the first three stages (*Figure 3—figure supplement 2*).

Finally, the GWAS and QTL peaks of association are situated in scaffold 419 of the WW reference assembly, which in a WW linkage map forms a linkage group along with six more scaffolds (472, 487, 515, 531, 540, and 609). We found that 12 genes present in this linkage group were differentially expressed, including *valkea* and *yellow-e* (*Supplementary file 1C*), and identified their orthologues in *D. melanogaster*.

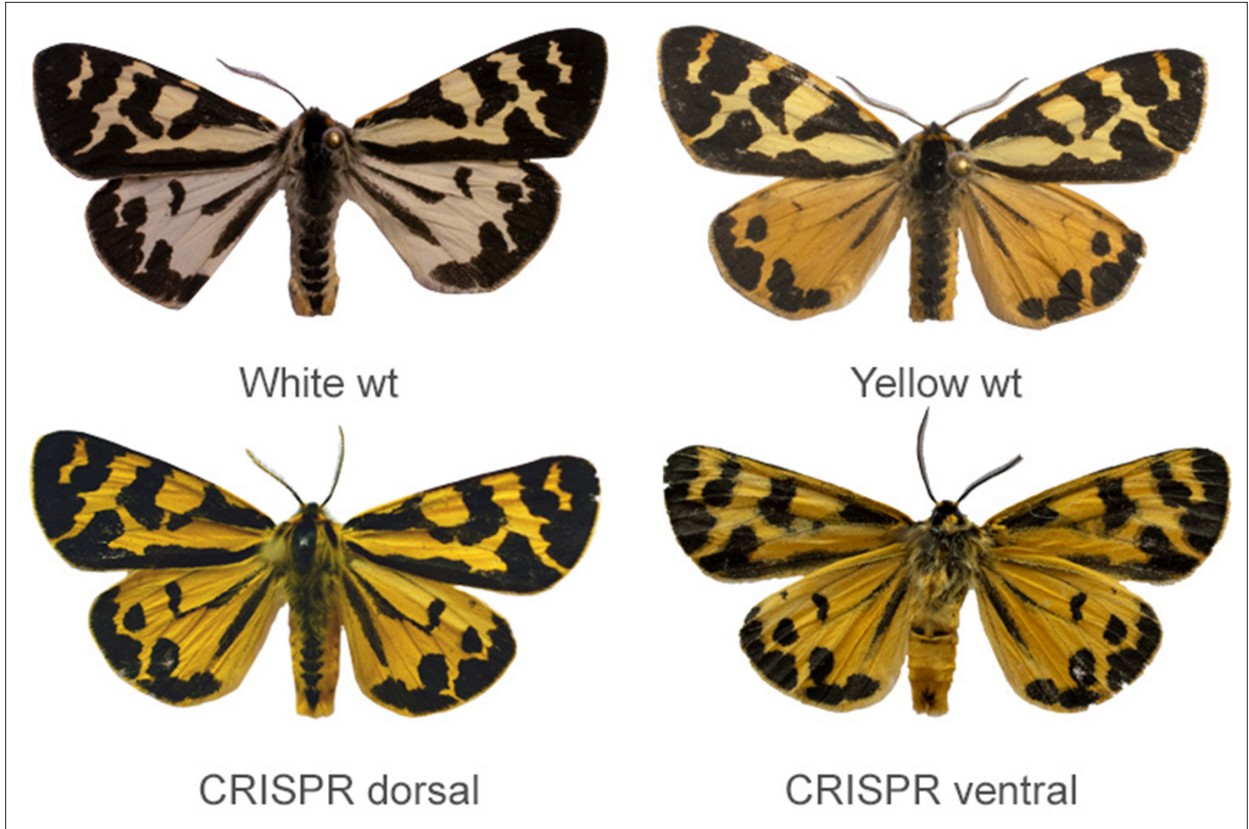

**Figure 4.** CRISPR/Cas9 knockouts of *valkea* transforms white scales into yellow scales across both hindwings and forewings. Wildtype WW and yy morphs (top), and the dorsal and ventral sides of one of the CRISPR knockout males (bottom).

The online version of this article includes the following figure supplement(s) for figure 4:

**Figure supplement 1.** The four males showing the mutant phenotype.

**Figure supplement 2.** Wildtype WW morphs show UV reflectance on the dorsal sides of the wings.

**Figure supplement 3.** CRISPR knockouts confirmed using sequence data of the five individuals with visible changes in phenotype.

**Figure supplement 4.** Reflectance spectra of the wings for wildtype Finnish genotypes (ww, wy and yy) and a yy Estonian.

## CRISPR/Cas9 knockouts of *valkea* produce yellow hindwings

To confirm the function of *valkea* in wing colouration, we used CRISPR/Cas9 to knock out the gene in white morphs. We tested five different guides to target the first three exons of *valkea*, and injected Cas9/sgRNA duplexes into a total of 1223 eggs. Of 143 larvae that hatched, only six developed to adults (*Supplementary file 1D*). However, of the five males that did eclose, four had a visible change in phenotype. Males produced yellow scales instead of white on the dorsal side of both the forewings and the hindwings (*Figure 4*). Forewings were more yellow than in the wildtype yellow males, which usually have lighter forewings compared to hindwings. White scales on the ventral side of the wings also became yellow, similar to wildtype yellows. Black melanin patterning did not seem to be affected. Variation in the amount of melanin can be attributed to the populations from which the individuals originated, with the darker samples coming from the Finnish population (*Figure 4—figure supplement 1*). Wildtype white morphs also reflect UV, particularly on the hindwings, but this is not seen in the CRISPR males (*Figure 4—figure supplement 2*). This could suggest a change in scale structure, or that the yellow pigment covers the UV-reflecting structures. To quantify changes in visible and UV colour, we took spectral measurements of the mutant males and compared them to wildtype males (*Figure 4—figure supplement 4*). The reflectance spectra for the hindwings of the CRISPR males most closely resembled that of wildtype yellow males, in both visible and UV wavelengths.

Four out of the five guides tested produced a mutant phenotype, with no differences in the male phenotype between guides. We used whole-genome sequences of the mutants to confirm that the

correct sites in *valkea* had been targeted (*Figure 4—figure supplement 3*). All samples also showed evidence of editing at the corresponding *yellow-e* exons, which mainly involved insertions. As all genotypes have similar forewing colour in the wildtypes, we do not expect *valkea* to affect the forewing and thus the change in forewing colour could be attributed to a *yellow-e* mutation. Only one female survived to adulthood, and this had a mosaic phenotype. Female colour does not correlate with the male colour genotypes, and the forewings of females are a pale-yellow colour. This individual with a mosaic phenotype had one mutant forewing which was much more yellow/orange than the wildtype. The rest of the wings and body resembled a wildtype female (*Figure 4—figure supplement 1/Figure 4—figure supplement 4*). Reflectance spectra show that the mutant left forewing is closer in colour to the yellow/orange on the hindwings, than to the colour of the opposite forewing (*Figure 4—figure supplement 4*). Since a *valkea* knockout is not expected to affect female phenotypes as they always have orange/red hindwings, this could be further evidence for the effect of *yellow-e* on forewing colour. We also checked for potential off-target effects of the CRISPR on other *yellow* genes. There was no evidence of editing (insertions, deletions or mutations) in the *yellow* genes *c, d2, f, f2, g2, h,* and *yellow* itself.

Survival of the eggs varied between the guides, although this was largely affected by the female, as hatching rate between females ranged from 0 to 70%. Females often lay unfertilised eggs, so we expect that hatching rate will be low in some crosses. Using two guides in combination did not produce any pupae or adults.

## Pigment analysis

Since the *yellow* gene family, to which *valkea* is related, is known to be responsible for the production of melanin pigments, we further investigated the identity of the wing pigmentation. First, we ruled out the presence of several non-melanin pigment types in the hindwings, including pterins and carotenoids. Pterins are commonly found in insects and, along with purine derivatives, papiliochromes and flavonoids, are soluble in strong acids and bases or in organic solvents (*Umebachi, 1975*; *Kayser, 1985*; *Shamim et al., 2014*). We placed wing samples from each morph in NaOH overnight, then measured the absorbance of the supernatant using a spectrophotometer. We also left wings in methanol overnight before measuring the supernatant. The spectra did not show any peaks indicative of any pigment dissolved in the sample. Similarly, we found no evidence for carotenoid pigments after dissolving in a hexane:tert-butyl methyl ether solution (*Figure 5—figure supplement 1*). Wings did not fluoresce under UV light, providing further evidence for the lack of fluorescent pigments including pterins, flavonoids, flavins, and papiliochromes (*Umebachi, 1975*; *Kayser, 1985*). Ommochromes are red and yellow pigments; high-performance liquid chromatography (HPLC) ruled out the presence of these pigments on the moth wings, which we compared to data from ommochrome-containing *Heliconius* wings and a xanthurenic acid standard (*Figure 5—figure supplement 2*).

HPLC analysis showed peaks characteristic of pheomelanin (*Figure 5*). Pheomelanins produce red-brown colour in grasshoppers and wasps (*Galván et al., 2015*; *Jorge García et al., 2016*), and orange-red colours in ants and bumblebees (*Hines et al., 2017*; *Polidori et al., 2017*). Insects generally have dopamine-derived pheomelanin and a breakdown product of this is 4-amino-3-hydroxyphenylethylamine (4-AHPEA) (*Barek et al., 2018*). Yellow wings showed large peaks for 4-AHPEA. White wings had around 27% of the 4-AHPEA levels seen in yellow wings, and black sections of the wings had 16%. Hydrogen iodide hydrolysis of wings produced the isomer 3-AHPEA, which may come from 3-nitrotyramine originating from the decarboxylation of 3-nitrotyrosine. Reduction of 3-nitrotyrosine produces 3-AHP, another marker of pheomelanin (*Wakamatsu et al., 2002*).

Analysis of the black portions of the wing found pyrrole-2,3-dicarboxylic acid (PDCA) and pyrrole-2,3,5-tricarboxylic acid (PTCA) (*Figure 5—figure supplement 3*). Both are components of eumelanin (*Barek et al., 2018*), suggesting that the black colouration seen in the wood tiger moth is predominantly eumelanin derived from dopamine. This is common in producing black colouration and providing structural components of the exoskeleton. In addition, dopamine is acylated to both *N*-β-alanyldopamine (NBAD) and *N*-acetyldopamine (NADA) sclerotins. NADA sclerotins are colourless and likely to be present on the white wings. This analysis of pigmentation is therefore consistent with a role for *yellow* family genes in regulating the colour polymorphism.

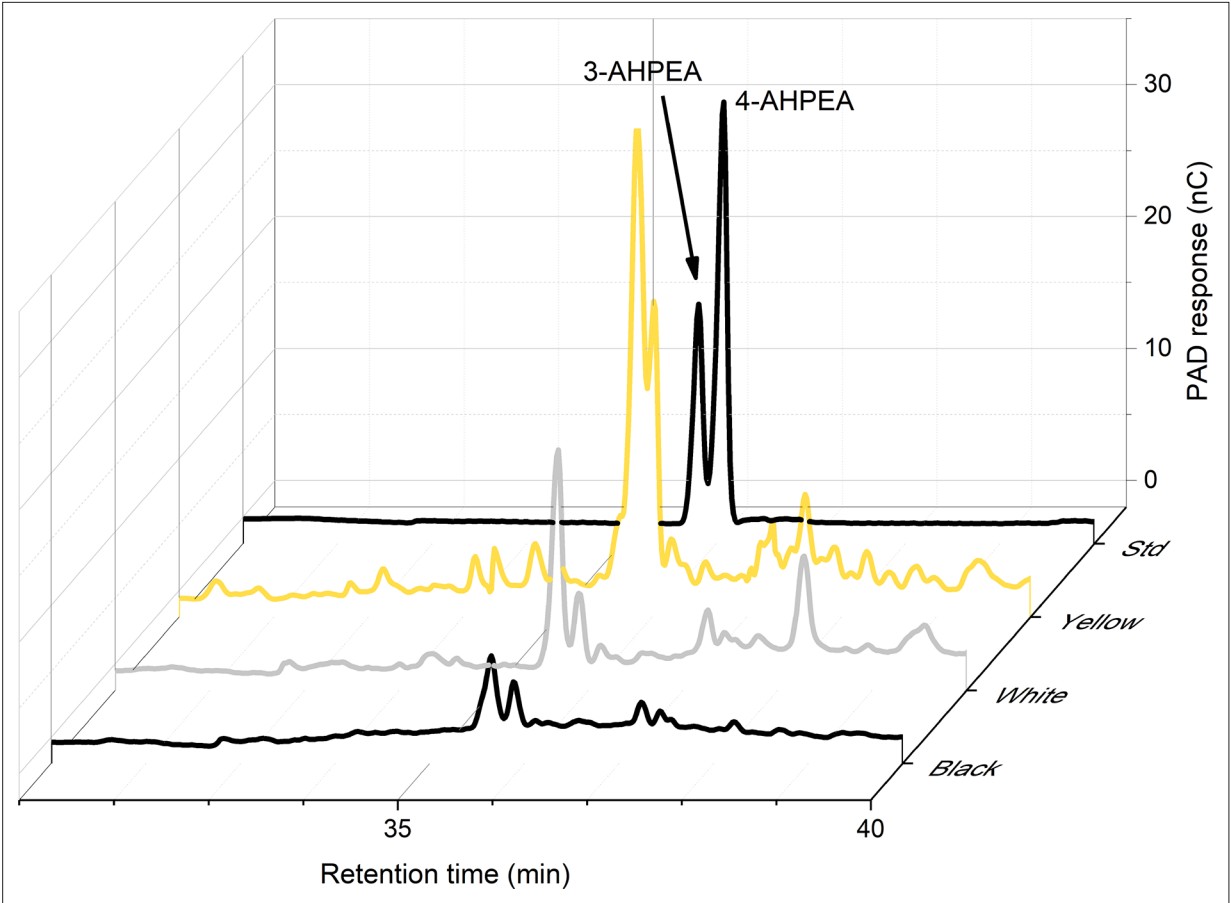

**Figure 5.** High-performance liquid chromatography (HPLC) analysis shows that the highest levels of 4-AHPEA, a breakdown product of pheomelanin, are seen in the yellow wings. Measurements for yellow, white, and black portions of the hindwing, plus the standard (Std) are shown.

The online version of this article includes the following figure supplement(s) for figure 5:

**Figure supplement 1.** Absorbance curves for yellow and white male wings and female wings left in (**A**) methanol, which absorbs at 205 nm, (**B**) sodium hydroxide, which absorbs in the UV range, and (**C**) hexane:tert-butyl methyl ether which absorbs between 195 and 210 nm, as shown in the control measurement. Peaks outside of these values would suggest the presence of pigment compounds in the solvent, but these are not seen.

**Figure supplement 2.** Spectral and chromatogram data obtained using high-performance liquid chromatography (HPLC) to test for the presence of ommochrome pigments.

**Figure supplement 3.** High-performance liquid chromatography (HPLC) result for eumelanin analysis.

## Discussion

Hindwing colouration of male *Arctia plantaginis* is polymorphic and these colour morphs vary in multiple behavioural and life-history traits, providing an example of a complex polymorphism. Here, we have shown that variation in male hindwing colour is associated with a duplicated sequence found only in white morphs and containing a gene from the *yellow* gene family. The white-specific copy, *valkea,* is highly expressed during pupal development, consistent with genetic dominance of the white allele. When *valkea* is knocked out in the white morphs, yellow pigment is produced, although due to the similarity of the sequences, *yellow-e* was also edited. While we cannot confirm that *valkea* is solely responsible for the white/yellow switch, we can rule out the role of other *yellow* family genes found along the same chromosome (*b*, *d2*, *h*, and *g2*).

These results add to the increasing evidence for the role of gene duplications in the evolution of adaptive genetic variation. Genes for the metabolism of proteins in *Heliconius* butterflies underwent several duplications, facilitating changes in diet and adaptation to pollen feeding (***Smith et al., 2016***). In *Zerene cesoina* butterflies, recent partial duplications of the transcription factor *doublesex*, resulting from multiple duplication events, are associated with sex-specific wing patterning. The duplicated

paralog acts as a repressor of genes producing UV-reflecting wing scales in females (*Rodriguez-Caro et al., 2021*).

We hypothesise that the morph-specific duplication that we see in *A. plantaginis* provides a region of reduced recombination between morphs, as the duplicated region is effectively hemizygous and cannot recombine except in homozygote genotypes, which could contribute to the maintenance of the complex polymorphism and the linkage of multiple traits. This is similar to the genetic architecture of the *Primula* supergene controlling heterostyly, which involves a large duplication containing five genes (*Huu et al., 2020*). In polymorphic *Papilio dardanus*, one colour pattern morph is associated with a duplicated region, again providing physical constraints on recombination (*Timmermans et al., 2014*). Nonetheless, in the case of the wood tiger moth, it remains unclear how a single gene, such as *valkea*, can control the development of a broad array of phenotypic traits.

One possible mechanism is that there is a regulatory element along the scaffold which is controlling colour via the *valkea* gene, but also regulating other genes to control different phenotypic traits. We found the most significant markers and SNPs located in a non-coding region close to the *yellow* genes, which likely contains a *cis*-regulatory element (CRE) controlling transcription of *valkea*. In cichlids, for example, a CRE at the gene encoding agouti-related peptide 2 controls variation in strip patterning in two closely related species (*Kratochwil et al., 2018*). Conserved CREs were shown to have wide-ranging effects on wing patterning in multiple Nymphalidae butterflies (*Mazo-Vargas et al., 2022*).

Differential expression of other genes on the same chromosome controlled by the CRE could explain variation in covarying traits. The overexpression of another gene, possibly encoding a zinc transcription factor, in yellow individuals in the early pupal stages suggests that there is differential expression of unlinked genes as a result of the polymorphism, although since this gene is on a different chromosome to *valkea* it is unlikely to be directly controlled by the CRE. Another hypothesis is that somehow *valkea* itself regulates other genes. However, *yellow* family genes are not known to regulate transcription of other genes, unlike, for example, *doublesex*, which undergoes alternative splicing and female mimetic wing pattern polymorphism in *Papilio polytes* (*Kunte et al., 2014*; *Nishikawa et al., 2015*).

The *yellow* family genes are highly conserved throughout insects (*Ferguson et al., 2011*). They have been widely linked to colouration (*Wittkopp et al., 2002*; *Miyazaki et al., 2014*; *Zhang et al., 2017a*; *Zhang et al., 2017b*), as well as behaviour, sex-specific phenotypes, and reproductive maturation (*Wilson et al., 1976*). These genes share a common origin with the major royal jelly protein (MRJP) genes (*Drapeau et al., 2006*) which are crucial in caste development in honeybees. Like the MRJP genes, *yellow* genes in honeybees have diverse spatial and temporal expression patterns. As our focus in *A. plantaginis* has been on wing tissue, we are missing expression of genes in other tissues that could be linked to other traits. Thus, it is not impossible to imagine that a *yellow* gene could have a similar function to a MRJP in regulating the development of a complex phenotype. Recent work with *Bicyclus anynana* showed that *yellow* functions as a repressor of male courtship (*Connahs, 2022*). On the other hand, sex-specific behavioural phenotypes of *yellow* mutants in *Drosophila* were found to be due to pigmentation effects (*Massey et al., 2019*), so more evidence is needed to suggest a functional role for *yellow* genes outside of pigmentation.

The duplication of *yellow-e* and surrounding regions in the white morphs suggests that the yellow morph is the ancestral form. *Valkea* could have evolved in a stepwise fashion, first as a tandem duplication then with a stop codon mutation altering the gene structure. Gene duplications can facilitate adaptation and, in some examples, lead to polymorphism. The fact that the white allele is dominant also supports the hypothesis that yellow is ancestral. Such invasions of new adaptive alleles are facilitated when the new allele is dominant, as it is then also expressed when heterozygous, that is, the Haldane's sieve effect. Melanism, for example, has repeatedly evolved in mammals due to dominant and semidominant mutations in the *Mc1r* locus which have become fixed (*Hoekstra, 2006*).

*Valkea* could represent an example of neofunctionalisation, where the duplicated gene gains a different function to the original gene copy. In the CRISPR mutants, both forewings and hindwings became yellow, and thus we hypothesise that *valkea* is controlling hindwing colour while *yellow-e* controls forewing colour. Since we do not expect *valkea* to have an effect on female wing colour, the change in forewing colour in the female could be attributed to the *yellow-e* knockout, although knockouts of *yellow-e* only are needed to confirm this. Those with the mutant phenotype showed only small deletions or insertions around the target site. By combining multiple guides we may expect to

see larger deletions (*Mazo-Vargas et al., 2022*); however, none of the eggs that were injected with more than one guide survived past the larval stage, suggesting that large deletions in *yellow* genes reduce fitness.

Contrary to previous work that attributed red and yellow colours to pterins in another tiger moth species (*Gawne and Frederik Nijhout, 2019*), we found high levels of 4-AHPEA in the yellow wings confirming the presence of pheomelanins. These pigments have been widely associated with red and yellow colours in mammals (e.g. *Mcgraw and Wakamatsu, 2004*), but only relatively recently described in insects and likely to be more widespread than previously thought. Yellow colours can also be produced by NBAD sclerotins which are sclerotising precursor molecules made from dopamine and these have an important role in the sclerotisation pathway for hardening the insect cuticle (*Andersen, 2007*; *Barek et al., 2017*) before becoming involved in melanisation (*Barek et al., 2018*). Thus, we suggest that the yellow colour arises partly from the NBAD sclerotins and partly from the presence of pheomelanin pigments, which has been proposed in other Lepidoptera (*Matsuoka and Monteiro, 2018*). While some 4-AHPEA also occurred in white wings, this may be due to its role in production of cross-linking cuticular proteins and chitin during sclerotisation (*Sugumaran, 2010*). Upregulation of genes on the white allele could be acting as a repressor of the generation of yellow colour. If we may speculate, *valkea* could impact the catalysis of dopamine, having cascading effects down the pathway resulting in the lack of yellow pigmentation. We suspect that *yellow* family genes play multiple roles within the melanin production pathway. In the wood tiger moth, *yellow* affects the conversion of DOPA into black dopamine melanin (*Galarza, 2021*). *Yellow-e* in particular has been linked to larval colouration in *B. mori* (*Ito et al., 2010*) and adult colour in beetles (*Wang et al., 2022*), while another gene, *yellow-f*, has a role in eumelanin production (*Barek et al., 2018*).

In summary, we identified a structural variant which is only present in white morphs of *A. plantaginis*. This region contains a previously undescribed gene, *valkea,* which when knocked out results in yellow wings. The presence of a regulatory element controlling wing colour and other traits via multiple downstream effects could explain how multiple traits are linked to wing colouration. This complex polymorphism allows multiple beneficial phenotypes to be inherited together, whereas recombination would separate multiple loci leading to maladapted individuals. Our results provide the basis for further exploration of the genetic basis of covarying behavioural and life-history traits, and offer an intriguing example for the role of gene duplications in adaptive variation.

## Methods
### Sampling
Homozygous lines of white (WW) and yellow (yy) *A. plantaginis* moths were created from Finnish populations at the University of Jyväskylä, Finland. Larvae were fed with wild dandelion (*Taraxacum* sp.) and reared under natural light conditions, with an average day temperature of 25°C and night temperature between 15 and 20°C. For the crosses, a heterozygous male, created from crossing a heterozygous male with a homozygous yy female, was backcrossed with a yy female. This was repeated to obtain four families totalling 172 offspring and 8 parents (*Supplementary file 1E*). Samples from wild populations were caught in Southern Finland (n = 10) and Central Finland (n = 20), where male morphs are either white or yellow, Estonia (n = 4), where males are mostly white, and Scotland (n = 4), where males are yellow (*Supplementary file 1F*). In addition, we included eight samples which are F1 offspring of wild Scottish samples. Forty pupae with known genotypes from lab populations (20 WW and 20 yy) were used for the RNA extractions.

### DNA extraction and sequencing
For the lab crosses, DNA was extracted from two legs crushed with sterilised PVC pestles using a QIAGEN DNeasy Blood & Tissue kit, following the manufacturer's instructions. Library preparation and GBS sequencing were performed by BGI Genomics on an Illumina HiSeq X Ten. For the wild samples, DNA was extracted from the thoraces also with a QIAGEN kit. Library preparation and sequencing were performed by Novogene (Hong Kong, China). 150 bp paired-end reads were sequenced on an Illumina NovaSeq 6000 platform.

## Linkage mapping analysis

FASTQ reads were mapped using bowtie v2.3.2 (*Langmead and Salzberg, 2012*) to the yellow *A. plantaginis* scaffold-level genome assembly (*Yen et al., 2020*). BAM files were sorted and indexed using SAMtools v.1.9 (*Li et al., 2009*) and duplicates removed using PicardTools MarkDuplicates (RRID:SCR_006525). Twelve samples which had aligned <30% were removed. Reads of the remaining samples had an average alignment of 94%. SNPs were called using SAMtools mpileup with minimum mapping quality set to 20 and bcftools call function. Lep-MAP3 (*Rastas, 2017*) was used for linkage map construction and we ran the following modules: ParentCall2 which called 105,622 markers, Filtering2, SeparateChromosomes2 with lodLimit = 5 and sizeLimit = 100, JoinSingles2All and OrderMarkers2 with recombination2 = 0 to denote the lack of female recombination. Genotypes were phased using the map2genotypes.awk script included with Lep-MAP3. Markers were named based on the genomic positions of the SNPs in the reference genome and the map. awk script, and this was used to further order the markers within the linkage groups. This resolved 30 linkage groups. Although we expect that there are 31 chromosomes in the moth genome, we suspect that the sex chromosome is missing in this assembly as the yy individual used in the genome assembly was female (*Yen et al., 2020*). A small number of markers which caused long gaps at the beginning or end of linkage groups were manually removed, leaving the final map 948.7 cM long with 19,803 markers. Markers were well distributed so we began the first analyses with this map. A linkage map was also assembled using sequences aligned to the white reference and this separated into 31 linkage groups.

The QTL analysis was carried out in R/qtl (*Broman et al., 2003*). Genotype probabilities were calculated before running a genome scan using the *scanone* function with the Haley–Knott method and binary model parameters, and including family as an additive covariate. The phenotype was labelled as either 0 (Wy) or 1 (yy). We ran 5000 permutations to determine the significance level for the QTL LOD scores. The *bayesint* function calculated the 95% Bayesian confidence intervals around the significant marker.

## Analysis of whole-genome sequences

FASTQ reads were mapped to the yellow *A. plantaginis* genome assembly (*Yen et al., 2020*) using BWA-MEM v7.17 (*Li, 2013*). As before, BAM files were sorted and indexed, and duplicates were removed. Genotyping and variant calling was carried out with the Genome Analysis Toolkit (GATK) (*McKenna et al., 2010*). Variants were called using HaplotypeCaller (v.3.7) in GVCF mode, then gVCFs combined with GenomicsDBImport (v.4.0). Joint genotyping was run with GenotypeGVCFs, set with a heterozygosity of 0.01, and SNPs were called using SelectVariants. Finally, the set of 20,787,772 raw SNPs were filtered using VariantFiltration and thresholds: quality by depth (QD > 2.0), root mean square mapping quality (MQ > 50.0), mapping quality rank sum test (MQRankSum > −12.5), read position rank sum test (ReadPosRankSum > −8.0), Fisher strand bias (FS < 60.0), and strand odds ratio (SOR < 3.0). A set of 5,227,288 SNPs passed the filtering.

We carried out a GWAS using the R package GenABEL v.1.8 (*Aulchenko et al., 2007*). The set of filtered SNPs were converted to BED format with PLINK2, keeping only biallelic SNPs (https://www.cog-genomics.org/plink/2.0/). Sites which were not in Hardy–Weinberg equilibrium (p<0.01), or had a call rate of <0.5, were excluded. Following this, 381,266 sites were retained across 40 individuals (out of 57). To account for population stratification, we performed MDS on kinship and identity-by-state (IBS) information estimated from the data, and included this as a covariate in the association test. Significance levels were calculated using Bonferroni corrected thresholds to account for multiple testing. Central and Southern Finnish populations were pooled for this analysis, based on a previous principal component analysis of these samples (*Yen et al., 2020*).

In *Yen et al., 2020*, many of these samples were processed in the same way but aligned to the white genome assembly. Read depth of the W-mapped samples was calculated in 1 kb windows across the candidate region using BEDtools (v.2.20.1) multicov (*Quinlan and Hall, 2010*). For visualisation, lines were smoothed using LOESS and span = 0.01 within ggplot2.

For analysis of structural variants, sequences from the white and yellow genome assemblies were aligned using MAFFT v7.450 (*Katoh and Standley, 2013*) and viewed with Geneious. Our focal sequence, scaffold 419 in the white genome, is the reverse complement of scaffold 206 in the yellow genome. *Figure 2C* was plotted with pafr (*Winter et al., 2020*).

## Identification of candidate genes and tree construction

To identify candidate genes in the QTL interval and GWAS region, we ran a protein BLASTP v.2.4.0 search to identify *H. melpomene* (Hmel2.5) proteins homologous to predicted *A. plantaginis* proteins in the region from the genome annotation. Informative gene names were obtained by performing a BLASTP search with the *H. melpomene* proteins against all *D. melanogaster* proteins in FlyBase v.FB2020_01 (flybase.org/blast).

For the *yellow* gene tree, we used Lepbase (*Challi et al., 2016*) to search for *yellow* genes in *B. mori* (ASM15162v1). We identified *yellow-e* in *H. melpomene* by searching for major royal jelly proteins, then comparing protein sequences of these against *Drosophila* proteins in FlyBase. The sequence for *Dmel yellow-e* was downloaded from FlyBase. To make the tree, coding sequences of all genes were aligned in Geneious using MAFFT v7.450 (*Katoh and Standley, 2013*), then the tree was constructed with PhyML using 10 bootstraps (*Guindon et al., 2010*).

## Differential gene expression

We dissected the wings out of the pupae in Cambridge, UK. Pupae and larvae were sent to Cambridge from Jyväskylä and were kept between 22 and 30°C. Pupae were sexed and only males were used. Dissections were made at four different stages: 72 hr after pupation (72 hr), 5 d after pupation (5 d; counting 0–24 first hours after pupation as day 1), pre-melanin deposition (Pre-mel), and post-melanin deposition (Mel). We sampled five individuals per stage and genotype. Hindwings and fore-wings were stored separately in RNA-later (Sigma-Aldrich) at 4°C for 2 wk and later transferred to –20°C, while the rest of the body was stored in pure ethanol. Only hindwings were used for RNAseq analysis.

Total RNA was extracted from hindwing tissue using a standard hybrid protocol. First, we transferred the wing tissue into Trizol Reagent (Invitrogen) and homogenised it using dounce tissue grinders (Sigma-Aldrich). Then, we performed a chloroform phase extraction, followed by DNase treatment (Ambion) for 30 min at 37°C. We measured the concentration of total RNA using Qubit Fluorometric Quantitation (Thermo Fisher) and performed a quality check using an Agilent 4200 TapeStation (Agilent). The extracted total RNA was stored at –20°C before being sent to Novogene UK for sequencing. Each individual was sequenced separately, with a total of 40 individual samples sequenced (five individuals per stage and genotype).

We performed quality control and low-quality base and adapter trimming of the sequence data using *TrimGalore!* We then mapped the trimmed reads to the two *A. plantaginis* genomes using STAR (*Dobin et al., 2013*). We performed a second round of mapping (2pass) including as input the output splice junctions from the first round. The *A. plantaginis* genome annotations WW and YY were included in each round of mapping respectively. We then used *FeatureCounts* to count the mapped reads. Finally, we used DESeq2 to analyse the counts and perform the DE analysis.

To identify the gene or genes controlling the development of wing colour in *A. plantaginis*, we performed a genome-wide differential expression analysis using limma-voom (*Ritchie et al., 2015*). First, we defined a categorical variable, 'GenStage', with eight levels containing the genotype and stage information of every individual sample (e.g. YY72h, WW72h, YY5days, etc.). Then, we built the design matrix fitting a model with GenStage as the only fixed effect factor contributing to the variance in gene expression and included family as a random effect factor (gene expression ~ 0 + GenStage + (1|Family)). We then filtered lowly expressed genes using the filterByExpr function in limma, which resulted in a reduction of the number of tested genes from 17,615–11,330 genes in the Y-mapped analysis and 17,930–10,920 in the W-mapped one. Then, we normalised the expression of the genes using the calcNormFactors function with TMM normalisation in limma and fit the design matrix using the voom function. We built a contrast matrix including the comparisons of interest, in which we compared the expression of the genotypes in each stage (i.e. h72 = WW72h-YY72h, d5 = WWd5-YYd5, Premel = WWPremel-YYPremel, Mel = WWMel-YYMel), and fit the contrast matrix to the data using the contrasts.fit function. Finally, we used the eBayer function on the fit dataset and we extracted the list of genes that are differentially expressed in each stage using the Benjamini–Hochberg procedure to correct for multiple testing. We evaluated the genome-wide gene expression using MDS using the plotMDS function of the limma package.

## Orthology assignment

To infer genome-wide orthology between *A. plantaginis* and *D. melanogaster*, we used OrthoFinder (v2.5.4) (*Emms and Kelly, 2019*). We used proteomes from six Lepidoptera species, *Plutela xylostella* (GCA_905116875_2), *B. mori* (GCF_014905235_1), *Spodoptera frugiperda* (GCF_011064685_1), *Parnassius apollo* (GCA_907164705_1), *Pieris macdunnoughi* (GCA_905332375_1), *Pararge aegeria* (GCF_905163445_1), and *D. melanogaster* (GCF_000001215_4). We ran the primary_transcript.py utility from OrthoFinder to extract only one transcript per protein, and then ran OrthoFinder with default settings.

## CRISPR/Cas9 genome editing

Guide RNAs were designed within the first three exons of *valkea* in the white genome annotation using Geneious (v. 2022.1.1). Guides were chosen with minimal off-target effects, high activity scores, and high specificity scores based on the Geneious algorithm (*Supplementary file 1G*). Guides in the first two exons of *valkea* showed off-target sites in *yellow-e*; however, those in exon 3 showed no off-target sites. Guide RNAs were synthesised by Sigma-Aldrich. Moths from the greenhouse populations, originating from Finnish and Estonian populations, at the University of Helsinki, Finland, were genotyped using DNA extracted from leg tissue using the Chemagic DNA tissue kit (Chemagen) and the primers detailed below. They were paired and left to mate overnight. Females were watched over the next 3–4 d for egg laying, and the eggs were removed and injected less than 6 hours after laying. Eggs were glued to microscope slides and injected with a 1:1 sgRNA/Cas9 mix with phenol red dye using pulled borosilicate glass capillaries. The injection mix contained 1 ug/ul Cas9, 500 ng/ul sgRNA, and 0.5% phenol red. Guides and Cas9 were diluted using low concentration TE buffer. Different combinations of guides were also injected in some samples, in which case these were mixed in a 1:1 ratio. Injections were performed using a MPPI-3 pressure injector with back pressure unit (ASI). In total, we injected 1223 eggs from 18 [WW × WW] or [WW × Wy] crosses. Larvae were kept individually in Petri dishes and fed daily with dandelion leaves. After eclosion, legs were taken from adults for DNA extraction. Library preparation and whole-genome sequencing (using Illumina NovaSeq 6000) of five CRISPR mutants were performed by CeGaT (Tübingen, Germany). We used these whole-genome sequences to confirm the editing of the *valkea* gene in the mutants. Sequences were aligned to the white reference genome using BWA-MEM as detailed earlier. We visualised the *valkea* gene sequences using Geneious and looked for insertions and deletions within and around the locations of the guides. This was repeated for the other *yellow* family genes (*b, c, d2, e, f, f2, g2, h*, and *yellow*).

## Genotyping white and yellow alleles

We used Primer3 to design primers within the duplicated region. Primers were expected to only amplify in WW and Wy individuals. The alignment of the white and yellow sequences was then used to design primers for genotyping the locus (*Supplementary file 1B*). We looked for short insertions or deletions that were fixed between the WW and yy within the *valkea/yellow-e* region, and put primers around these structural variants. Primers were tested on DNA extractions from moths of known genotypes, including both sexes, wild and lab samples. We used Sanger sequencing of the PCR product to confirm the correct sequences were amplified. A set of primers successfully amplified a 449 bp region downstream of *valkea* within the duplication. This amplified in WW and Wy samples, but not in yy (*Figure 2—figure supplement 4—source data 1*). We found that white alleles have a 35 bp deletion within an intron of the *yellow-e* gene. We amplified a 163 bp region around this (YY_ tarseq_206_arrow:9,846,212–9,846,375) using a standard PCR protocol which allowed us to identify the allele based on the size of the PCR product. Yellow alleles produce the full 163 bp sequence, while white alleles produce a smaller 128 bp product (*Figure 2—figure supplement 4*). Heterozygotes have a copy of each and show both bands on a gel.

## Photography and spectrophotometry

Photographs of CRISPR mutant and wildtype moths were taken under standard lighting conditions with a Samsung NX1000 digital camera converted to full-spectrum with no quartz filter to enable ultraviolet (UV) sensitivity fitted with a Nikon 80 mm lens. A UV and infrared blocking filter was used for the human-visible photos, which transmits wavelengths between 400and 680 nm (Baader UV/IR Cut Filter). For the UV images, a UV pass filter was used (Baader U filter), which transmits wavelengths

between 320 and 380 nm. Images were standardised using grey-scale reflectance standards (Avian Technologies, Micro FSS08).

Reflectance spectra of coloured regions of the forewings and hindwings of 22 lab stock (including WW, Wy, and yy genotypes) and five CRISPR mutant moths were recorded with a UV-VIS spectrometer (Ocean Insight HR4PRO) connected to a xenon light source (Ocean Insight PX-2). Measurements were normalised using a diffuse white standard (Spectralon 99%). We used the OceanView software (v.2.0.8) to record scans with a boxcar width of 5 and integration time of 5000 ms. Measurements were repeated three times and the mean used. Reflectance spectra were plotted and analysed using the R package *pavo* (*Maia et al., 2013*).

## Pigment analysis

### Solubility and fluorescence tests

Five hindwings from each morph were placed in two separate solvents (0.1 M NaOH and 90% MeOH) and left overnight. The supernatant was analysed with an Agilent Cary 8454 UV-Visible spectrophotometer and the spectra compared to known spectra for various pigments. A UV lamp (Philips TL8W/08F8T5/BLB) was used to test for fluorescence on the wings. The presence of carotenoids was tested by placing wings into 1 ml of pyridine and leaving at 95°C for 4 hr (*McGraw et al., 2002*). To these we added 1 ml of 1:1 hexane:tert-butyl methyl ether and 2 ml of water before shaking and leaving overnight. Again, the supernatant was measured with the spectrometer.

### HPLC test for eumelanin and ommochrome pigments

To determine the type of melanin producing the black colour on the wings, we cut out approximately 5 mg of the black sections of the wings, from both females and males. Eumelanin analysis was carried out according to *Borges et al., 2001*. Each sample was added to a tube containing 820 μl 0.5 M NaOH, 80 μl 3% $H_2O_2$ and an internal standard (48 nmol phthalic acid) and heated in a boiling water bath for 20 min. Once cool, 20 μl of 10% $Na_2SO_3$ and 250 μl of 6 M HCl were added. Samples were then extracted twice with 7 ml of ethyl acetate. Ethyl acetate was dried at 45°C under a stream of nitrogen. The residue was dissolved into 0.5 ml of 0.1% formic acid.

We carried out HPLC on an Agilent 1100 HPLC. 20 μl of the sample was injected into a Waters Atlantis T3, 100 × 3.0 mm i.d. analytical column (Waters, Milford, MA). The column was set to 25°C and analytes were detected at wavelength 280 nm. The HPLC mobile phase consisted of two eluents: UHQ-water/MeOH (98/2; v/v) with 0.1% formic acid and UHQ-water/MeOH (40/60; v/v) with 0.1% formic acid. Flow rate was 0.4 ml/min and the used gradient started with 100% of eluent A and ramped evenly from time 0–15 min to 40:60 (A:B; v/v), held at 40:60 for 6 min, and ramped evenly back to initial eluent composition (100% A) over 5 min. We compared chromatograms obtained from the samples to the chromatograms obtained from synthetic melanin, ink from sepia officinalis and black human hair.

HPLC was also applied to observe the possible presence of ommochrome pigments. Injection volume was 10 μl and for the separation we used the same Waters Atlantis T3 column (100 × 3.0 mm i.d.) set to 30°C. Solvent A was UHQ-water and B was acetonitrile (ACN), both containing 0.1% formic acid. Flow rate was 0.4 ml/min and the used gradient was as follows: initial flow ratio was 98/2 water/ ACN (v/v) ramping then evenly from time 1–15 min to 30:70 water:ACN (v/v), held for 1.5 min and then ramped evenly back to initial eluent composition over 0.5 min. The column was stabilised for 7 min before a new run.

### Pheomelanin analysis

Samples were analysed for pheomelanin content according to the method of *Kolb et al., 1997* with modifications. A 2 mg sample was placed in a screw-capped tube with 100 μl water, 500 μl ~55–58% hydrogen iodide (HI), and 20 μl 50% hypophosphorous acid ($H_3PO_2$). Samples were capped tightly and hydrolysed for 20 hr at 130°. After cooling, samples were evaporated under nitrogen flow, then dissolved in 1 ml of 0.1 M HCl and purified with solid-phase extraction. Strata SCX cartridges were preconditioned with 2 ml of methanol, 3 ml of water, and 1 ml of 0.1 M HCl. Sample was then applied to the cartridge, washed with 1 ml of 0.1 M HCl, and finally eluted with 1 ml of methanol (MeOH): 0.5 M ammonium acetate ($NH_4CH_3CO_2$) (20:80 v/v).

Hydrogen iodide hydrolysis products were determined by a Dionex HPLC equipped with pulsed amperometric detection (HPLC/PAD). A Phenomenex Kinetex C18 column (150 × 4.6 mm i.d.; 5 μm particle size) with a gradient elution (*Supplementary file 1H*) at a flow rate of 0.9 ml min$^{-1}$ with the eluents: (A) sodium citrate buffer (*Hines et al., 2017*) in ultra-high-quality water (internal resistance ≥ 18.2 MΩ cm; Milli-Q Plus; Millipore, Bedford, MA) and (B) methanol were used for the separation. Dionex ED-50 pulsed amperometric detector (Dionex, Sunnyvale, CA) equipped with a disposable working electrode by using a Dionex waveform A with potentials presented in *Supplementary file 1I* was used for detection. The preparation method for the 4-AHP, 3-AHPEA, and 4-AHPEA standards used in calibration is described in *Wakamatsu et al., 2014*.

## Acknowledgements

We thank Alma Oksanen, Kaisa Suisto, and the greenhouse staff for insect rearing, and Elisa Salmivirta and Sari Viinikainen for lab assistance. Thanks to Emeritus Prof. Shosuke Ito for kindly providing pheomelanin standards, Bodo Wilts for advice on the pigment analyses, and James Barnett for help with the spectrophotometry. We thank Muktai Kuwalekar, Claudius Kratochwil, Rachel Blow, Ian Warren, Tom Generalovic, and Joe Hanly for providing equipment and advice regarding the CRISPR injections. Funding This work was supported by the Academy of Finland grants to MB (#343356) and JM (projects 345091 and 328474), and a Biotechnology and Biological Sciences Research Council (BBSRC) grant to CJ (046_BB_V0145X_1).

## Additional information

### Funding

| Funder | Grant reference number | Author |
| --- | --- | --- |
| Academy of Finland | 343356 | Melanie N Brien |
| Academy of Finland | 345091 | Johanna Mappes |
| Academy of Finland | 328474 | Johanna Mappes |
| Biotechnology and Biological Sciences Research Council | 046_BB_V0145X_1 | Chris D Jiggins |

The funders had no role in study design, data collection and interpretation, or the decision to submit the work for publication.

### Author contributions

Melanie N Brien, Data curation, Formal analysis, Funding acquisition, Investigation, Visualization, Writing – original draft, Writing – review and editing; Anna Orteu, Formal analysis, Investigation, Visualization, Writing – original draft, Writing – review and editing; Eugenie C Yen, Investigation, Writing – review and editing; Juan A Galarza, Resources, Data curation, Writing – review and editing; Jimi Kirvesoja, Investigation, Writing – original draft, Writing – review and editing; Hannu Pakkanen, Resources, Investigation, Methodology, Writing – review and editing; Kazumasa Wakamatsu, Resources, Methodology, Writing – review and editing; Chris D Jiggins, Johanna Mappes, Conceptualization, Resources, Supervision, Funding acquisition, Writing – review and editing

### Author ORCIDs

Melanie N Brien ⓘ https://orcid.org/0000-0002-3089-4776
Hannu Pakkanen ⓘ http://orcid.org/0000-0002-8725-1931
Kazumasa Wakamatsu ⓘ http://orcid.org/0000-0003-1748-9001
Chris D Jiggins ⓘ https://orcid.org/0000-0002-7809-062X

### Decision letter and Author response

Decision letter https://doi.org/10.7554/eLife.80116.sa1
Author response https://doi.org/10.7554/eLife.80116.sa2

## Additional files

### Supplementary files

• Supplementary file 1. Supplementary tables. (A) 21 genes are within in the QTL interval. Start and end positions shown are on scaffold 206 in the yellow *A. plantaginis* reference genome. Gene sequences were blasted against *Heliconius melpomene* and searched for in FlyBase. Apla gene names from annotations produced by *Yen et al., 2020*. (B) Primers used for genotyping. Tested using GoTaq Flexi buffer and GoTaq DNA polymerase, with annealing temperature of 57°C for 35 cycles. 'Ye12' primers surround a small deletion in white alleles, producing a 163 bp product from Y alleles and 128 bp product from W alleles. 'Dup5' primers amplify a 449 bp sequence within the duplicated sequence only in moths with at least one W allele. See *Figure 2—figure supplement 4* for gel images. (C) List of differentially expressed genes found in the linkage group containing scaffold 419 (WW reference). (D) Details of the number of eggs injected with each sgRNA and those which produced adult moths. (E) Sample list of all lab cross individuals used in linkage mapping. (F) Sample list of all wild samples used. (G) CRISPR sgRNAs sequences. (H) Elution gradient used in pheomelanin HPLC analysis. (I). Waveform of disposable working electrode in pheomelanin analysis.

• MDAR checklist

### Data availability

Scripts and data for the QTL, GWAS and DE analyses can be found at doi: https://doi.org/10.5281/zenodo.8208751. RADseq, RNAseq data, and WGS of CRISPR samples were deposited to SRA under study accession number PRJNA937225. Raw sequencing data of wild samples has previously been deposited in ENA, study accession No. PRJEB36595.

The following datasets were generated:

| Author(s) | Year | Dataset title | Dataset URL | Database and Identifier |
|---|---|---|---|---|
| Brien MN, Orteu A | 2022 | Colour polymorphism associated with a gene duplication in male wood tiger moths | https://doi.org/10.5281/zenodo.8208751 | Zenodo, 10.5281/zenodo.8208751 |
| Brien MN, Orteu A | 2022 | Arctia plantaginis Raw sequence reads | https://www.ncbi.nlm.nih.gov/bioproject/PRJNA937225/ | NCBI BioProject, PRJNA937225 |

The following previously published dataset was used:

| Author(s) | Year | Dataset title | Dataset URL | Database and Identifier |
|---|---|---|---|---|
| Yen EC | 2020 | A haplotype-resolved, de novo genome assembly for the wood tiger moth (Arctia plantaginis) through trio binning | https://www.ebi.ac.uk/ena/browser/view/PRJEB36595 | EBI European Nucleotide Archive, PRJEB36595 |

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
