## [Editor Report]

Through genetic mapping and analysis of WGS data, the authors identify a gene duplication co-segregating with a color polymorphism in males of the aposematic tiger moth. They name the new gene valkea and investigate its expression and function in relation to wing pigmentation. Using CRISPR to disrupt valkea, they observe changes in wing color. However, because valkea was not the only gene edited, its causal role in the color polymorphism cannot be unambiguously established.

---

## [Decision Letter]

**Decision letter after peer review:**

Thank you for submitting your article "Colour polymorphism associated with a gene duplication in male wood tiger moths" for consideration by *eLife*. Your article has been reviewed by three peer reviewers, and the evaluation has been overseen by a Reviewing Editor and Christian Landry as the Senior Editor. The reviewers have opted to remain anonymous.

The reviewers provided very thorough individual reviews and have also validated each other's comments. The Reviewing Editor has subsequently drafted this to help you prepare a revised submission.

Essential revisions:

1) To frame it in terms of "supergene entailing reduced recombination", the work requires quantification of "lower recombination" within the duplicated segment, and more detailed characterization of the 5' end of that segment. Alternatively, claims of "supergene"-like behavior should be explicitly stated as a hypothesis. In terms of "supergene" pleiotropic effects, it seems that the association between duplication and polymorphism is shown directly only for pigmentation, and not any other phenotypes that covary with that. The association to other traits should also be presented as hypothetical.

2) Definite proof that valkea, and not something else in the duplicated region (e.g. regulatory sequence responsible for expression differences between morphs for other genes in that linkage group), is responsible for the white phenotype requires functional analysis. Possibly, the more accessible type of analysis would involve using CRISPR-Cas9 to knock-out valkea from a white morph background. That being impossible, showing spatial patterns of valkea (and other genes in that linkage group?) expression (e.g. using in situ hybridization) in developing wings of the white morph would at least already associate valkea to that specific region of the wing and add support to it being involved in the COLOR (not scale maturation, for example) polymorphism.

3) Provide more details on the methods, including making replication and data structure clearer in the gene expression analysis (and plotting actual data points in Figure 2B).

*Reviewer #1 (Recommendations for the authors):*

Line:125-126. "likely to be mapping errors". What do the authors mean by 'mapping errors'? greater specificity is needed. Importantly, I would like to see some attempt to document what you think is going on. If you filter your mapping using MAPQ > 30, when mapping across to the entire genome, does this region lose more reads in the yellow samples than what you show in Figure 2? Do all of the while individuals show this higher coverage, compared to yellow? Not clear in Figure 2 if the read depth here is the total for all of the individuals in your collection. Did you look at other regional samples that you sequenced?

The genomic region flanking valkea is not very well characterized in the manuscript. Figure 2 is only showing a cartoon, while there are perfectly good methods for aligning these two regions and showing computationally inferred orthology for this region. More specifically, while the downstream region of yellow-g, yellow-e both look orthologous, the upstream region appears to have different loci (ie. jg6744, jg1307). This suggests that this simplified cartoon is masking a lot more complexity, and I am asking for that to be presented clearly and empirically, as this is currently … glossed over/ignored in the relevant section of the results (lines:112-126).

Lines:150-151: I can understand your reasoning, but this is because I understand quite a bit about the temporal dynamics of color deposition in Lepidoptera wings. Most readers will not. Please provide more of your reasoning here, in terms of thinking that this color change is not due to patterning genes (though nearly all, or all?, aforementioned genes associated with Lep wing color changes, as not associated with color biosynthesis genes, but regulatory/patterning genes). So, your logical step here is quite a departure from the literature, please justify edifying the reader.

Gene expression patterns. I greatly appreciate that you provide an overview via a PCA-like plot to see the clustering of your samples. But.. Figure S3: is this an MDS plot (as per Edger), or something else? You do not describe how you generated this figure in the methods and that should be clarified. Also, in the relevant main text, lines: 160-161, you make a very qualitative statement, and I can't tell if that's just the authors "eye-balling" the PCA-like plot.

Since the RNAseq analysis was working with WW vs. yy individuals, how do the authors envision the expression threshold of valkea to give rise to a dominant phenotype? Stated another way, if white individuals still arise from Wy males, and in those the expression of valkea is going to be much lower … how do they envision the functioning of their new gene in a heterozygous background giving rise to a binary trait?

Figures. I was surprised to see that none of the figures had a general header before the subpanels were described (i.e. a one-sentence overview). I find this very strange and suggest the authors do this.

Figure 3 could benefit from more clarity. A is I guess a restructuring of all your RNAseq data to only look at differences between the two color morphs only, grouping all tissues together? This was not really clear in the main text and is not clarified here. I assume B is only looking at valkea expression across all time points … but this should be made clear.

Line 179: this analysis is fine, but I am rather unhappy with calling this pooling of all tissues and looking for only morph differences, as 'genome-wide analysis of RNAseq' … as all of your analyses are looking at RNAseq data mapped to the genome.. there is nothing unique here compared to what was done previously, expect that tissues are pooled by morph -- but this is not described clearly in the methods (lines: 447-457). Please, revise your methods for greater clarity of your two-step approach, and revise your main text, and figure legends accordingly. Perhaps more importantly, what do you gain by doing this two-step approach? I can see the logic, that even with this type of dev stage grouping, valkea clearly an outlier. This perspective should be shared with the reader. Having that come before the tissue-specific result works, but currently, you present the tissue-specific, then the pooled tissue, and then the figure panels are in the wrong order … it could be more linear and clear. Please revise.

Topology approach. This section appears rather rushed and should be introduced with greater clarity for the reader. Also, why are you only doing this for such a narrow region of the chromosome? Why are you not doing this for the whole region flanking the valkea insertion region? Where is the actual location of yellow-e in this figure? Again, it brings up the strange part this manuscript, in that the authors appear again to be avoiding their 5' flanking region of the duplication … why? That should be mirroring this pattern, which would strengthen the message here, but it is not presented. In sum, one can only really appreciate S5 if you can see the larger region, the flanking loci, the repeated patterns, and some proper phylogeny explaining the alternative topologies (as I find the text description alone lacking proper clarity for the topology alternatives). Does this arise due to the low coverage of your individual WGS data?

Recombination. I find it rather strange that you discuss the potential for recombination suppression as a result of the duplication, yet conduct no measures of LD. Why? You have many whole-genome datasets from a sufficient number of individuals for some preliminary analyses at least, to provide quantitative evidence. But, upon closer reading, is this because you have too little depth per individual for this? This brings up the issue that average read depth per individual is not clearly reported, and that needs to be changed in the main text.

Where is the table of the data generated per individual, for RAD and WGS? Their genomic coverage after mapping? In the area of the text where I expected this, I found instead % of reads mapping.. that doesn't convey depth, which conveys accuracy of WGS data … please make a table for these standard metrics common to QTL and GWAS papers.

*Reviewer #2 (Recommendations for the authors):*

This study truly is a fantastic effort to identify the locus responsible for adaptive color polymorphism in tiger moths. In general, the paper is well-written and the figures communicate the main results quite well. Following are suggestions, concerns, and/or questions I have about the study that I believe could improve the study and paper.

As mentioned in the public review, I have concerns with the hypotheses the authors use to frame the paper. I see this study as a quite well-executed effort to identify the genetic and phenotypic basis of wing color polymorphism in these tiger moths. I do clearly see how the study was designed to distinguish between the involvement of "large structural variants" versus "sing gene mutations". I think this could be addressed through some revisions in the Introduction. Along the same lines, I don't see any need to introduce the concept of supergenes, as I don't see any efforts to directly test if a co-adapted gene complex is involved. Again, this can be addressed through limited text editing.

This study would be greatly strengthened by additional gene expression and/or functional data. Spatial expression data of valkea and yellow-e in developing hindwings could provide critical evidence of these genes involved in the color pattern differences. Such data has been critical in the implication of other color pattern genes involved in Heliconius and Bicyclus wing development. Even further, functional confirmation, through methods such as CRISPR-cas9 editing has proven to be extremely successful to confirm the role of candidate genes in butterfly wing pattern development ( see examples from Heliconius, Bicyclus, Colias, and other butterflies), including successful CRISPR edits of yellow to study gene function in other butterfly species. Recent other studies of butterfly color pattern genetics published in *eLife* have included such spatial expression data and/or functional data. I remain unconvinced from the tree topology analyses that valkea alone at this locus is involved in generating the color differences, or that valkea acts as the genetic switch for the color polymorphism. To find the results of this study as convincing as those other recent studies, I would need to see comparable evidence.

For the pigment analyses, after the pheomelanin is extracted from yellow wings, do the wings appear white instead of yellow? I would be curious to see an image of what the extracted wings looked like, so I could directly connect the HPLC differences with a change in yellow versus white coloration.

I feel the paper could be strengthened through some integration of the genetic and phenotypic results. The authors have a rich RNA-seq dataset that can be used to characterize clusters and networks of genes expressed in development, and differences between the color morphs. There is also a well-resolved melanin pathway, with some knowledge of specific gene functions from *Drosophila* and other butterfly studies. In this regard, I feel the authors have missed an opportunity to integrate their gene expression data with their phenotypic data. For instance, what other genes do valkea and yellow-e cluster with (e.g. show correlated expression pattern with) in the RNA-seq data? These clusters would reflect the network of genes that are differently expressed between color morphs. I would in interested in knowing what these genes are and if there are any genes with interesting functions or known to be in developmental pathways that involve yellow genes, or are involved in pigmentation. In the melanic pathway, it could be powerful to visualize where in the pathway the authors propose that valkea may be impacting pheomelanin production. I would urge the authors to revisit Matsuda and Monteiro 2020 as an example of how such data can be integrated to give the reader a more clear and integrated understanding of how the genetic changes identified may be impacting the phenotype.

I quite like that the authors highlight gene duplication as a structural variant that is largely unable to properly recombine with haplotypes lacking the duplicated region. I would urge the authors to cite other examples where such duplications have been implicated in wing pattern development and adaptive evolution. For example, gene duplicates have been implicated in the adaptive evolution of pollen feeding in Helcinius butterflies (Smith et al. 2020) and sexually dimorphic color pattern development in Zerene butterflies (Rodriguez et al. 2021). This paper has an opportunity to highlight the increasing evidence of recent gene duplications in evolutionary diversification.

The duplicated region at the mapped locus needs to be further resolved. At a minimum, the authors should finely annotate the duplicated region. For instance, are there any TE insertions? Are the entire duplicate regions reflect a single recent duplication? Or, are there regions duplicated more than once, and this region appears to have experienced several instances of unequal crossovers and potential insertion/deletion events? Is the regulatory region (e.g. 5' UTR, etc.) duplicated? Does the regulatory region show elevated divergence relative to the other duplicated regions?

Similarly, further analysis of valkea would strengthen the paper. Does valkea show any evidence of adaptive molecular evolution? Are there non-synonymous substitutions with yellow-e? How old/recent is the gene duplication event?

Further analyses to address these questions could provide further resolution to the evolution and potential role of valkea in the color polymorphism.

Figure 2D. I have some reservations on interpreting the read-coverage as evidence the duplicated region is missing in all yellow samples. For instance, yellow-g shows a similar mapped reads pattern as the region just 3' of valkea in the duplicated region, yet yellow-g is not considered to be within the duplicated region. Are the regions in the duplicated region with high coverage for yellow samples potentially repetitive regions of the genome, such as TEs? If so, an annotation of this region would improve our ability to interpret the read coverage results.

Also, did the authors attempt to map RNA-seq reads from yellow individuals to a white reference genome to see if any reads mapped to valkea? This would be a quick and direct way to confirm that valkea is not present/expressed in any yellow genomes. In the methods section, it does not state which A. plantaginis genome the RNA-seq gata was mapped to. If RNA-seq data for yellow individuals was only mapped to a yellow reference genome that lacks valkea, then we can not be sure if valkea transcripts are actually absent from yellow RNA-seq samples (I honestly assume the authors are aware of the bias introduced by mapping yellow RNA-seq data to a yellow reference genome only, but I just need to check since I couldn't discern from the methods).

*Reviewer #3 (Recommendations for the authors):*

Specific comments to the authors:

Line 26: the limitation of recombination does not necessarily imply a supergene architecture. Furthermore, your results point a pleiotropic effect of a single gene rather than to a combined effect of several genes, therefore departing from the classical 'supergene' hypothesis. I would recommend rephrasing this part.

Line 40: it is unclear to me what you mean by 'selection is context-dependent, this needs to be explained in more detail.

Line 49: in mimetic butterflies, there is also a series of inversions at the supergene controlling colour pattern polymorphism in H. numata (Jay et al. 2021 Nature Genetics).

Line 59: it is unclear what you mean by 'in an ecological context', you may explain the key ecological features involved in the persistence of the polymorphism in this species.

Line 70: What is causing the mating advantage? Is it linked to female preference? If so, this raises the question of the selection promoting the evolution of such preference?

Figure 1: it this the frequency of MALE colour patterns shown on panel A?

Line 131: In my opinion figure S2 should be in the main document, it is very important to infer the ancestral state and the origin of the duplicated region. I would prefer moving panel D of figure 2 into the supplementary if space is missing.

In figure 2 panel D, I guess you compared YY HOMOZYGOUS males with WY HETEROZYGOUS males? This would be useful to provide this genotypic information in the legend.

Line 148: you may be precise that the RNAseq was performed on the wing disk. Did you investigate the expression patterns in hindwings and forewings separately? This might be interesting since the level of yellow colour seem to be higher in the hindwing than in the forewings (at least from what I can see in figure 1).

Line164: This suggests that there is not major shift in expression patterns between morphs even within the wing disk tissue. This is in apparent contradiction with the 99 DE genes found at the genomic level (lines 180-181). I think I misunderstood something here, these first expression analyses were restricted to genes located within the QTL region? This should be clarified.

Line 170-171: Did the overexpression of yellow-e occur at the same developmental stage as the overexpression of valkea (i.e. premelanin stage)? This is important to infer the putative developmental pathway inducing white colour pattern development.

Figure S5: The position of the yellow-e gene and of the valkea gene are not indicated in the figure, so it is difficult to draw conclusions from this figure at this point.

Line 196: This provides quite indirect evidence for ruling out the effect of yellow-e on the switch between white and yellow colour pattern development. The overexpression of yellow-e at the pre-melanin stage could be caused by variation in the (non-coding) regulatory region, and therefore explaining why variation in the yellow-e sequences is not specifically associated with colour pattern variation.

Line 291: In line with your conclusions, the dominance of the 'white' allele over the 'yellow' one is consistent with the white allele being a derived haplotype that invaded an ancestrally yellow population. Such invasion of a new adaptive allele is facilitated when the invading allele is dominant over the ancestral one because it is then expressed at a heterozygous state (i.e. Haldane's sieve effect).

Line 297: I have some trouble reconciling the 'neofunctionalization hypothesis' with the fact that valkea seems to be a truncated gene. Is there any example where a truncated yellow gene gained a new function in the melanin developmental pathway?

The overexpression of the valkea gene could stem from a lack of regulation of a gene with a loss of function. In that case, the switch in colour pattern might stem from variation in the non-coding region affecting the expression of other genes, like yellow-e. Is there a way you can rule out this alternative hypothesis?

[Editors’ note: further revisions were suggested prior to acceptance, as described below.]

Thank you for resubmitting your work entitled "Colour polymorphism associated with a gene duplication in male wood tiger moths" for further consideration by *eLife*. Your revised article has been evaluated by Christian Landry (Senior Editor) and a Reviewing Editor.

The manuscript has been improved but there are some remaining issues that need to be addressed, as outlined below:

The CRISPR experiment is important but lacks a more detailed description, as well as earlier and more explicit acknowledgement of its limitations, including that it failed to conclusively demonstrate that valkea (and not yellow-e) is responsible for the white/yellow switch. This uncertainty should be referred to earlier on (abstract?).

Relative to standard butterfly color pattern analysis, more information is necessary regarding the UV analysis (methods and wildtype phenotype), and regarding the use of "eumelanin" and "pheomelanin" which are usually reserved for vertebrates.

*Reviewer #2 (Recommendations for the authors):*

I have reviewed the revisions, and the authors have sufficiently addressed my previous concerns and suggestions. However, the authors' inclusion of additional CRISPR data is lacking critical information and analyses, which I detail below.

Lines 217 and 218 states that whole genome sequences of mutants were used to confirm mutants. However, there is no description of the methods used, nor can I find that those data are made available. Please add a description of the methods used for whole genome sequencing and confirming the presence of mutant alleles. I am also interested in what methods were used to test for off-target effects. It is particularly important to examine for potential off-target edits to other yellow genes.

Ln 220. Only one female survived to adulthood, and this had a mosaic phenotype. "This individual had one yellow forewing, similar to the male mutants, with the rest of the body and wings being wildtype (Figure 4 —figure supplement 1)." It is not at all clear that this female has one mutant wing. Both wings appear much more yellow than a white wildtype. I need some further phenotypic evidence (spectrophotometer readings or pigment analyses) as the phenotypic variation is not evident in the images provided. It would be ideal to see that the colors in mosaic mutant phenotypic regions are significantly different from wildtype (this can be done using spec readings from multiple wildtype wings and mutant wings). Second, there needs to be sequence verification of the mutations included in the manuscript, as previously mentioned.

Figure 4 —figure supplement 2 shows images UV. However, there are no methods provided for how these UV data were collected. Without some details of the imaging setup, I am unable to discern that images reflect differences in UV reflection, or may be due to variations in the imaging procedure. If possible, spectra analyses of the wings are an easy and cost-effective approach to quickly confirming changes in UV brightness on lepidoptera wings.

There is also no background information given for the wildtype UV. Lines 212-213 suggest the UV is a result of scale structures. What is the reference or evidence for this? Variation in UV reflection is known to be influenced by pigment composition in Pieris butterflies, not necessarily scale structures. To make assertions of UV being associated with scale structures, I would be interested in seeing the characterization of the putative UV related scale structures in wildtypes and mutants. This type of scale characterization (e.g. SEM and/TEM of wing scales in wildtype and mutants) is routinely included with other functional genomic studies of similar wing colorations (for examples see Ficarrotta et al. 2022, Livraghi et al. 2022, Concha et al. 2019, Matsuoka and Monteiro 2018). At a minimum, a detailed description/characterization of the wildtype UV should be given to the readers. Along these lines, I am curious to know if the UV may be iridescent. If so, some descriptive info on the iridescence would be needed (i.e. angle of incidence). Also, if iridescent, the differences in UV between wildtype and mutants should be examined further to determine if the image differences between wt and mutants are due to changes in the angle of incidence.

Ln 335-336. I am unclear what evidence supports yellow-e having a forewing-specific effect.

Ln 337-338 The authors state that yellow-e was "likely also knocked out…". I think this is misleading, as lines 218-219 states "All samples also showed evidence of editing at the corresponding yellow-e exons, which mainly involved insertions". Based on this it seems more than "likely", and actually confirmed yellow-e coding was disrupted in ALL samples.

*Reviewer #3 (Recommendations for the authors):*

The revised version of the manuscript successfully addresses most of my previous concerns.

Results from CrispR/cas9 experiments targeting the valkea gene were added to the manuscript in order to validate the role of this gene in the developmental switch from the yellow to the white morph. Such CrispR/cas9 experiments are challenging and obtaining high number of mutant adults is usually difficult in Lepidoptera.

Here a few male mutants and one female mutant were successfully obtained. Nevertheless, the lack of specificity of the CrispR guides resulted in modifications in both valkea and yellow-e genes in the few mutant individuals that reached the adult stage, therefore preventing the full characterisation of the respective functional implications of these two genes in the development of hind and forewing colour patterns in males and females.

From what I understood, the main argument for ruling out yellow-e as causing the white/yellow switch in male hindwings is the phenotype observed in a single mutant female showing in panel E of the supplementary figure 4. The sentences line 221-224 are not entirely convincing to me. The phenotype of the mutant female is used to point at the putative role of yellow-e on forewing colour in female. Does it lead to hypothesize a role of yellow-e on forewing colour in both sexes? And thus to a role of valkea in male hindwing colour? This indirect argument should be made clearer, and further discussion on the respective roles of these two genes in hind and forewing coloration is needed.

In the supplementary figure 4 (referred to as Figure 4 —figure supplement 1): the panel E shows the phenotype of the mutant female but picture of the wild-type female would be useful to fully evaluate the impact of the CrispR treatment on phenotypic variation.

Line 213: This is quite interesting, did you observe differences in scale structure between wild-type yellow and white scales, and in the wild-type yellow vs. mutant yellow scales? Such observations on the respective role of pigments and scale structure in the reflected colours are also relevant to understand the developmental bases of wing colour variations.

---

## [Author Response]

Essential revisions:1) To frame it in terms of "supergene entailing reduced recombination", the work requires quantification of "lower recombination" within the duplicated segment, and more detailed characterization of the 5' end of that segment. Alternatively, claims of "supergene"-like behavior should be explicitly stated as a hypothesis. In terms of "supergene" pleiotropic effects, it seems that the association between duplication and polymorphism is shown directly only for pigmentation, and not any other phenotypes that covary with that. The association to other traits should also be presented as hypothetical.2) Definite proof that valkea, and not something else in the duplicated region (e.g. regulatory sequence responsible for expression differences between morphs for other genes in that linkage group), is responsible for the white phenotype requires functional analysis. Possibly, the more accessible type of analysis would involve using CRISPR-Cas9 to knock-out valkea from a white morph background. That being impossible, showing spatial patterns of valkea (and other genes in that linkage group?) expression (e.g. using in situ hybridization) in developing wings of the white morph would at least already associate valkea to that specific region of the wing and add support to it being involved in the COLOR (not scale maturation, for example) polymorphism.3) Provide more details on the methods, including making replication and data structure clearer in the gene expression analysis (and plotting actual data points in Figure 2B).

We have made extensive revisions to this manuscript, the main addition being a CRISPR/Cas9 experiment in which we functionally validate the valkea gene. Regarding the essential revisions:

1. We have edited the text to move the focus away from supergenes and look more generally at the possible genetic basis of complex polymorphisms and adaptive variation, presenting supergenes as one possibility. The main figures now include more of the 5’ region of the duplication which shows the sequence upstream of the duplication is highly similar in the white and yellow genomes (Figure 2C). In the discussion, we hypothesise that this morph-specific duplication will provide a region of reduced recombination because the region is effectively hemizygous and cannot recombine except in homozygote genotypes. We cannot carry out a detailed LD analysis as the sequence is not present in both morphs. We have added PCR assays as extra evidence for the lack of the duplicated region in yellow morphs (Appendix figure 1). This genotyping assay is based on a small deletion in the yellow-e gene in white morphs.

2. We have added the results of a CRISPR gene editing experiment, in which we knocked out the valkea gene leading to the white morphs becoming yellow. We suspect that these results also show a role for yellow-e in forewing colouration, as sequencing of these individuals suggested mutations in both valkea and yellow-e due to the similarity of the sequences.

3. The methods for the gene expression analysis have been restructured and clarified.

We thank the three reviewers for their detailed comments and hope that the following changes fully address their questions and concerns. New and edited text is highlighted in blue on the revised manuscript.

Reviewer #1 (Recommendations for the authors):Line:125-126. "likely to be mapping errors". What do the authors mean by 'mapping errors'? greater specificity is needed. Importantly, I would like to see some attempt to document what you think is going on. If you filter your mapping using MAPQ > 30, when mapping across to the entire genome, does this region lose more reads in the yellow samples than what you show in Figure 2? Do all of the while individuals show this higher coverage, compared to yellow? Not clear in Figure 2 if the read depth here is the total for all of the individuals in your collection. Did you look at other regional samples that you sequenced?

We have taken a more detailed look at the mapping across the whole scaffold by including a comparison of different mapping quality filters. This shows that when MQ filters are more stringent, read depth decreases more in yellow samples compared to white in the duplication region (Figure 2 supplements 2 and 3). Thus we think this shows that in yellow individuals, yellow-e reads are mapping to valkea (and in white morphs vice versa). In general, MQ is lower in the duplication in both white and yellow samples.

The patterns of higher coverage in white individuals were also seen when including samples from the non-Finnish populations (Scotland and Estonia). Figure 2D shows the mean read depth of the white and yellow samples, and this has been clarified in the figure legend. None of the white samples had 0 coverage in the valkea region.

The genomic region flanking valkea is not very well characterized in the manuscript. Figure 2 is only showing a cartoon, while there are perfectly good methods for aligning these two regions and showing computationally inferred orthology for this region. More specifically, while the downstream region of yellow-g, yellow-e both look orthologous, the upstream region appears to have different loci (ie. jg6744, jg1307). This suggests that this simplified cartoon is masking a lot more complexity, and I am asking for that to be presented clearly and empirically, as this is currently … glossed over/ignored in the relevant section of the results (lines:112-126).

Figure 2C has been edited with the cartoon replaced with a more specific alignment of the two regions which shows clearly the missing region in the yellow genome. This also now includes more of the region upstream of the duplication. We have added an explanation that jg6744 and jg1307 are the same gene, and all genes in the flanking regions are present in both morphs (lines 146-149).

Lines:150-151: I can understand your reasoning, but this is because I understand quite a bit about the temporal dynamics of color deposition in Lepidoptera wings. Most readers will not. Please provide more of your reasoning here, in terms of thinking that this color change is not due to patterning genes (though nearly all, or all?, aforementioned genes associated with Lep wing color changes, as not associated with color biosynthesis genes, but regulatory/patterning genes). So, your logical step here is quite a departure from the literature, please justify edifying the reader.

It is confusing that we here mentioned genes associated with patterning rather than colour. Instead we have now included the example of melanin pathway genes from Zhang et al. 2017, which is a more relevant analysis of melanin production in Lepidoptera. We have also included a comparison of gene expression of yellow genes from Ferguson et al., 2011 (now lines 157-160).

Gene expression patterns. I greatly appreciate that you provide an overview via a PCA-like plot to see the clustering of your samples. But.. Figure S3: is this an MDS plot (as per Edger), or something else? You do not describe how you generated this figure in the methods and that should be clarified. Also, in the relevant main text, lines: 160-161, you make a very qualitative statement, and I can't tell if that's just the authors "eye-balling" the PCA-like plot.

Figure S3 (now Figure 3 supplement 1) is an MDS plot generated in limma and we added this information to the methods. We have edited the figure legend to include this and a better description of the plot. We did not formally test the effect of developmental stage in explaining most of the genome-wide variation. However, the lack of clustering among samples of the same morphs could be consistent with the fact that the phenotype is controlled by a single Mendelian locus and thus few genes are DE between morphs. These results suggest that more genes are involved in expression profiles specific to the developmental stage, rather than the wing phenotype. We have added this explanation to the text.

Since the RNAseq analysis was working with WW vs. yy individuals, how do the authors envision the expression threshold of valkea to give rise to a dominant phenotype? Stated another way, if white individuals still arise from Wy males, and in those the expression of valkea is going to be much lower … how do they envision the functioning of their new gene in a heterozygous background giving rise to a binary trait?

It is unknown if the expression of valkea is lower in Wy. The W allele is fully dominant over y, with heterozygotes presenting the dominant hindwing colour phenotype and not intermediate phenotypes. Thus we expect expression of valkea to be the similar in WW and Wy individuals, although there are differences in other traits between WW and Wy such as UV reflectance.

Figures. I was surprised to see that none of the figures had a general header before the subpanels were described (i.e. a one-sentence overview). I find this very strange and suggest the authors do this.

Figure legends have been edited.

Figure 3 could benefit from more clarity. A is I guess a restructuring of all your RNAseq data to only look at differences between the two color morphs only, grouping all tissues together? This was not really clear in the main text and is not clarified here. I assume B is only looking at valkea expression across all time points … but this should be made clear.

More details have been added to the figure legend to make these plots clearer. Figure 3A is showing only DE genes at the pre-melanin stage, while 3B is showing only expression of valkea across the different stages. Overall, the gene expression methods have been restructured into a more logical order for the reader.

Line 179: this analysis is fine, but I am rather unhappy with calling this pooling of all tissues and looking for only morph differences, as 'genome-wide analysis of RNAseq' … as all of your analyses are looking at RNAseq data mapped to the genome.. there is nothing unique here compared to what was done previously, expect that tissues are pooled by morph -- but this is not described clearly in the methods (lines: 447-457). Please, revise your methods for greater clarity of your two-step approach, and revise your main text, and figure legends accordingly. Perhaps more importantly, what do you gain by doing this two-step approach? I can see the logic, that even with this type of dev stage grouping, valkea clearly an outlier. This perspective should be shared with the reader. Having that come before the tissue-specific result works, but currently, you present the tissue-specific, then the pooled tissue, and then the figure panels are in the wrong order … it could be more linear and clear. Please revise.

We have modified the gene expression Results section and methods to clarify the strategy that was followed. Only one analysis was performed in which all genes of all samples were analysed. However, initially we wanted to specify that we had a list of 22 candidate genes found in the QTL/GWAS region that we were particularly interested in, as possible cis-regulatory changes found in the associated could be affecting gene expression of nearby genes.

Topology approach. This section appears rather rushed and should be introduced with greater clarity for the reader. Also, why are you only doing this for such a narrow region of the chromosome? Why are you not doing this for the whole region flanking the valkea insertion region? Where is the actual location of yellow-e in this figure?

It wasn’t clear here that we had excluded the duplication because the white and yellow morphs cannot be compared due to lack of sequence in yellows. Because we cannot make this comparison, we decide to remove this analysis, and it does not provide sufficient evidence for or against the role of yellow-e in hindwing colour.

Again, it brings up the strange part this manuscript, in that the authors appear again to be avoiding their 5' flanking region of the duplication … why? That should be mirroring this pattern, which would strengthen the message here, but it is not presented. In sum, one can only really appreciate S5 if you can see the larger region, the flanking loci, the repeated patterns, and some proper phylogeny explaining the alternative topologies (as I find the text description alone lacking proper clarity for the topology alternatives). Does this arise due to the low coverage of your individual WGS data?

Initially we did not include detailed analysis of the 5’ flanking region because this did not fall within the region which was found to be significant in the QTL analysis. The genes in this region are orthologous in the white and yellow reference genomes, and we do not see any additional large structural variation. This is now clearer in Figure 2C.

Recombination. I find it rather strange that you discuss the potential for recombination suppression as a result of the duplication, yet conduct no measures of LD. Why? You have many whole-genome datasets from a sufficient number of individuals for some preliminary analyses at least, to provide quantitative evidence. But, upon closer reading, is this because you have too little depth per individual for this?

We discuss recombination suppression as one potential consequence of the duplication. The absence of the duplicated region in the yy genome indicates that the sequence is effectively hemizygous and so we cannot determine if there is a change in LD across white and yellow morphs.

This brings up the issue that average read depth per individual is not clearly reported, and that needs to be changed in the main text.Where is the table of the data generated per individual, for RAD and WGS? Their genomic coverage after mapping? In the area of the text where I expected this, I found instead % of reads mapping.. that doesn't convey depth, which conveys accuracy of WGS data … please make a table for these standard metrics common to QTL and GWAS papers.

Summary statistics have been added to the supplementary tables (S5 and S6) for the QTL and GWAS samples.

Reviewer #2 (Recommendations for the authors):This study truly is a fantastic effort to identify the locus responsible for adaptive color polymorphism in tiger moths. In general, the paper is well-written and the figures communicate the main results quite well. Following are suggestions, concerns, and/or questions I have about the study that I believe could improve the study and paper.As mentioned in the public review, I have concerns with the hypotheses the authors use to frame the paper. I see this study as a quite well-executed effort to identify the genetic and phenotypic basis of wing color polymorphism in these tiger moths. I do clearly see how the study was designed to distinguish between the involvement of "large structural variants" versus "sing gene mutations". I think this could be addressed through some revisions in the Introduction. Along the same lines, I don't see any need to introduce the concept of supergenes, as I don't see any efforts to directly test if a co-adapted gene complex is involved. Again, this can be addressed through limited text editing.

We have edited both the introduction and discussion to move the focus away from supergenes, and instead base our hypotheses around the possible mechanisms for variation in colour polymorphisms and adaptive variation.

This study would be greatly strengthened by additional gene expression and/or functional data. Spatial expression data of valkea and yellow-e in developing hindwings could provide critical evidence of these genes involved in the color pattern differences. Such data has been critical in the implication of other color pattern genes involved in Heliconius and Bicyclus wing development. Even further, functional confirmation, through methods such as CRISPR-cas9 editing has proven to be extremely successful to confirm the role of candidate genes in butterfly wing pattern development ( see examples from Heliconius, Bicyclus, Colias, and other butterflies), including successful CRISPR edits of yellow to study gene function in other butterfly species. Recent other studies of butterfly color pattern genetics published in eLife have included such spatial expression data and/or functional data. I remain unconvinced from the tree topology analyses that valkea alone at this locus is involved in generating the color differences, or that valkea acts as the genetic switch for the color polymorphism. To find the results of this study as convincing as those other recent studies, I would need to see comparable evidence.

We have added the results of our gene editing experiment which successfully showed that when valkea is knocked out in a white morph, yellow pigment is produced on the wings. We remain unsure of the role of yellow-e but we use these results to hypothesise that valkea controls hindwing colour, while yellow-e affects forewing colour. Sequencing showed that both valkea and yellow-e had evidence of gene editing around the target guide sequences.

We agree that the tree topology did not provide any convincing evidence for or against the role of yellow-e vs. valkea, so as mentioned earlier, we decided to remove this analysis since we cannot estimate topology at the valkea gene.

For the pigment analyses, after the pheomelanin is extracted from yellow wings, do the wings appear white instead of yellow? I would be curious to see an image of what the extracted wings looked like, so I could directly connect the HPLC differences with a change in yellow versus white coloration.

Unfortunately the methods mean that this is not possible as the wings are crushed to extract the pheomelanin. The hydrogen iodide treatment is a harsh strong acid, leaving the residues of all samples a dark brown colour.

I feel the paper could be strengthened through some integration of the genetic and phenotypic results. The authors have a rich RNA-seq dataset that can be used to characterize clusters and networks of genes expressed in development, and differences between the color morphs. There is also a well-resolved melanin pathway, with some knowledge of specific gene functions from *Drosophila* and other butterfly studies. In this regard, I feel the authors have missed an opportunity to integrate their gene expression data with their phenotypic data. For instance, what other genes do valkea and yellow-e cluster with (e.g. show correlated expression pattern with) in the RNA-seq data? These clusters would reflect the network of genes that are differently expressed between color morphs. I would in interested in knowing what these genes are and if there are any genes with interesting functions or known to be in developmental pathways that involve yellow genes, or are involved in pigmentation. In the melanic pathway, it could be powerful to visualize where in the pathway the authors propose that valkea may be impacting pheomelanin production. I would urge the authors to revisit Matsuda and Monteiro 2020 as an example of how such data can be integrated to give the reader a more clear and integrated understanding of how the genetic changes identified may be impacting the phenotype.

This is a very good suggestion. We looked at the functions of other DE genes but none stood out as being part of melanin or pigmentation pathways. Valkea in the white genome sits next (5’) to yellow family genes g and e. yellow genes together with laccase2 lie upstream of the insect melanin pathway and act as master genes involved in the catalysis of dopa-melanin and dopamine-melanin to produce black and brown melanin respectively. The production of dopamine-melanin can be suppressed further down the pathway by the conversion of dopamine to N-b-alanyldopamine (NBAD) though the binding of β-alanine by the activity of ebony, forming NBAD, the precursor of yellow sclerotin resulting in a yellow pigment.

In recent insect studies (Galván et al., 2015; Jorge García et al., 2016; Matsuoka and Monteiro, 2018; Polidori et al., 2017; Zhang et al., 2019 doi:10.3390/ijms20112728), including ours, the yellow pigmentation is also attributed, at least partly, to pheomelanin derived by the oxidation of dopamine. Both the NBAD and pheomelanin routes are final molecule products of the melanin pathways. Hence, if we may speculate, valkea could impact the catalysis of dopamine having cascading effects down the pathway resulting in (lack of) yellow pigmentation. The precise interactions with other genes in the dopamine pathway are currently unknown. Because of these unknowns we have not added any further figures relating to the pigment analysis or melanin pathway (which is described in better detail in Matsuoka and Monteiro, as the reviewer suggests).

I quite like that the authors highlight gene duplication as a structural variant that is largely unable to properly recombine with haplotypes lacking the duplicated region. I would urge the authors to cite other examples where such duplications have been implicated in wing pattern development and adaptive evolution. For example, gene duplicates have been implicated in the adaptive evolution of pollen feeding in Helcinius butterflies (Smith et al. 2020) and sexually dimorphic color pattern development in Zerene butterflies (Rodriguez et al. 2021). This paper has an opportunity to highlight the increasing evidence of recent gene duplications in evolutionary diversification.

These are great examples and we have edited the discussion to include more examples for the role of gene duplications in adaptive variation.

The duplicated region at the mapped locus needs to be further resolved. At a minimum, the authors should finely annotate the duplicated region. For instance, are there any TE insertions? Are the entire duplicate regions reflect a single recent duplication? Or, are there regions duplicated more than once, and this region appears to have experienced several instances of unequal crossovers and potential insertion/deletion events? Is the regulatory region (e.g. 5' UTR, etc.) duplicated? Does the regulatory region show elevated divergence relative to the other duplicated regions?Similarly, further analysis of valkea would strengthen the paper. Does valkea show any evidence of adaptive molecular evolution? Are there non-synonymous substitutions with yellow-e? How old/recent is the gene duplication event?Further analyses to address these questions could provide further resolution to the evolution and potential role of valkea in the color polymorphism.

These are all interesting questions. We used RepeatMasker and found a number of TEs within the region. There are no TEs in the coding sequences of valkea, and the density of TEs in this region is not obviously different from surrounding regions. At the moment we can’t say much more about this so haven’t included any specific analysis in the manuscript.

From looking at whole scaffold alignments, we do not see evidence for further major duplications in this region.

We are lacking data that would allow us to estimate mutation rate and the age of the duplication event. From a previous study (Yen et al. 2020), we know that the population in Georgia is separated from the Finnish and Estonian populations. Georgian samples do not have the duplication suggesting a more recent timescale. Further analysis of valkea is planned with new data and we are looking to include this in future manuscripts.

Figure 2D. I have some reservations on interpreting the read-coverage as evidence the duplicated region is missing in all yellow samples. For instance, yellow-g shows a similar mapped reads pattern as the region just 3' of valkea in the duplicated region, yet yellow-g is not considered to be within the duplicated region. Are the regions in the duplicated region with high coverage for yellow samples potentially repetitive regions of the genome, such as TEs? If so, an annotation of this region would improve our ability to interpret the read coverage results.

We used PCR assays as additional evidence for the lack of valkea in yellow individuals. Although coverage of yellow-g does fluctuate across the gene, the pattern is fairly consistent in both whites and yellows. There is no pattern between the location of the TEs and the peaks in coverage.

Also, did the authors attempt to map RNA-seq reads from yellow individuals to a white reference genome to see if any reads mapped to valkea? This would be a quick and direct way to confirm that valkea is not present/expressed in any yellow genomes. In the methods section, it does not state which A. plantaginis genome the RNA-seq gata was mapped to. If RNA-seq data for yellow individuals was only mapped to a yellow reference genome that lacks valkea, then we can not be sure if valkea transcripts are actually absent from yellow RNA-seq samples (I honestly assume the authors are aware of the bias introduced by mapping yellow RNA-seq data to a yellow reference genome only, but I just need to check since I couldn't discern from the methods).

Yes, the results presented in the DE analysis section are using reads mapped to the white reference (line168).

Reviewer #3 (Recommendations for the authors):Specific comments to the authors:Line 26: the limitation of recombination does not necessarily imply a supergene architecture. Furthermore, your results point a pleiotropic effect of a single gene rather than to a combined effect of several genes, therefore departing from the classical 'supergene' hypothesis. I would recommend rephrasing this part.

We have rephrased the abstract, along with the general focus of the introduction and discussion, to move away from the supergene hypothesis and talk more generally about the genetic basis of polymorphisms in wild populations, including supergenes as one of several possible mechanisms.

Line 40: it is unclear to me what you mean by 'selection is context-dependent, this needs to be explained in more detail.

Here we are referring to genetic correlations between the colour locus and other traits which have benefits in different contexts, such as in the examples presented in the following paragraph, where white and yellow males have differences in traits which, for example, give them an advantage in mating or predator defence.

Line 49: in mimetic butterflies, there is also a series of inversions at the supergene controlling colour pattern polymorphism in H. numata (Jay et al. 2021 Nature Genetics).

A reference to the H. numata supergene has been added (line 51).

Line 59: it is unclear what you mean by 'in an ecological context', you may explain the key ecological features involved in the persistence of the polymorphism in this species.

There has been lots of research relating to the persistence of the polymorphism in this well-studied species. We have tried to make sure that the introduction covers the key studies while not becoming too detailed.

Line 70: What is causing the mating advantage? Is it linked to female preference? If so, this raises the question of the selection promoting the evolution of such preference?

This paragraph has been updated with the most recent studies in this area. While females generally prefer to mate with white males (Nokelainen et al., 2012), their preference is thought to be flexible and affected by male morph frequency, as males of either morph have a reproductive advantage when they are the most common morph (Gordon et al., 2015). Yellow males are generally less successful in their reproductive output (De Pasqual et al., 2022).

Figure 1: it this the frequency of MALE colour patterns shown on panel A?

Yes, this has been clarified.

Line 131: In my opinion figure S2 should be in the main document, it is very important to infer the ancestral state and the origin of the duplicated region. I would prefer moving panel D of figure 2 into the supplementary if space is missing.

After looking at various ways to present this section, we decided to keep this figure in the supplementary (now Figure 2 – supplement 4). However, we have added some further discussion about the possible ancestry of yellow-e (lines 328-333).

In figure 2 panel D, I guess you compared YY HOMOZYGOUS males with WY HETEROZYGOUS males? This would be useful to provide this genotypic information in the legend.

Yes, figure 2D is comparing wild white individuals which are likely a combination of WW and Wy, with wild yy. The figure legend has been edited.

Line 148: you may be precise that the RNAseq was performed on the wing disk. Did you investigate the expression patterns in hindwings and forewings separately? This might be interesting since the level of yellow colour seem to be higher in the hindwing than in the forewings (at least from what I can see in figure 1).

The RNAseq analysis used only hindwings dissected from pupae and we have clarified this in the methods.

Line164: This suggests that there is not major shift in expression patterns between morphs even within the wing disk tissue. This is in apparent contradiction with the 99 DE genes found at the genomic level (lines 180-181). I think I misunderstood something here, these first expression analyses were restricted to genes located within the QTL region? This should be clarified.

This has been clarified with the restructuring of the DE section. There are more genes that are DE between stages than between colour morphs, thus developmental stage drives the clustering.

Line 170-171: Did the overexpression of yellow-e occur at the same developmental stage as the overexpression of valkea (i.e. premelanin stage)? This is important to infer the putative developmental pathway inducing white colour pattern development.

Yes, these are both overexpressed in the pre-melanin stage. This has been added to line 184.

Figure S5: The position of the yellow-e gene and of the valkea gene are not indicated in the figure, so it is difficult to draw conclusions from this figure at this point.Line 196: This provides quite indirect evidence for ruling out the effect of yellow-e on the switch between white and yellow colour pattern development. The overexpression of yellow-e at the pre-melanin stage could be caused by variation in the (non-coding) regulatory region, and therefore explaining why variation in the yellow-e sequences is not specifically associated with colour pattern variation.

As mentioned earlier, we removed this figure and analysis because, as the reviewer suggests, it was difficult to draw any conclusions from this.

Line 291: In line with your conclusions, the dominance of the 'white' allele over the 'yellow' one is consistent with the white allele being a derived haplotype that invaded an ancestrally yellow population. Such invasion of a new adaptive allele is facilitated when the invading allele is dominant over the ancestral one because it is then expressed at a heterozygous state (i.e. Haldane's sieve effect).

This is a great point to include the Haldane’s sieve effect and we have added it to the discussion (lines 328-333).

Line 297: I have some trouble reconciling the 'neofunctionalization hypothesis' with the fact that valkea seems to be a truncated gene. Is there any example where a truncated yellow gene gained a new function in the melanin developmental pathway?The overexpression of the valkea gene could stem from a lack of regulation of a gene with a loss of function. In that case, the switch in colour pattern might stem from variation in the non-coding region affecting the expression of other genes, like yellow-e. Is there a way you can rule out this alternative hypothesis?

We believe that the CRISPR mutants provide evidence that valkea is having a direct effect on the phenotype. In the mutants, both forewings and hindwings became yellow, and thus we suspect that valkea is controlling hindwing colour while yellow-e, which was likely also knocked out due to the similarity of the sequences, controls forewing colour. However, because of this we cannot completely rule out the role of yellow-e in hindwing colouration.

[Editors’ note: what follows is the authors’ response to the second round of review.]

The manuscript has been improved but there are some remaining issues that need to be addressed, as outlined below:The CRISPR experiment is important but lacks a more detailed description, as well as earlier and more explicit acknowledgement of its limitations, including that it failed to conclusively demonstrate that valkea (and not yellow-e) is responsible for the white/yellow switch. This uncertainty should be referred to earlier on (abstract?).

We have made it clear in the abstract that both valkea and the original yellow-e gene were knocked out in our CRISPR experiment. We also added further clarification of this in the first paragraph of the discussion. Further discussion of the role of yellow-e remains later in the manuscript (lines 350-353).

Relative to standard butterfly color pattern analysis, more information is necessary regarding the UV analysis (methods and wildtype phenotype), and regarding the use of "eumelanin" and "pheomelanin" which are usually reserved for vertebrates.

Detailed methods for the UV photography have been added at lines 576-582. Further comparison of UV reflectance in wildtype vs. mutant phenotypes can be seen in the reflectance spectra (Figure 4 – supplement 4). The use of eumelanin and pheomelanin in insects is covered in the paper by Barek et al. 2018 (doi: 10.1111/pcmr.12672), which we cite in the pigment Results section and in the discussion. Other examples of pheomelanins in insects are at lines 260-262.

Reviewer #2 (Recommendations for the authors):I have reviewed the revisions, and the authors have sufficiently addressed my previous concerns and suggestions. However, the authors' inclusion of additional CRISPR data is lacking critical information and analyses, which I detail below.Lines 217 and 218 states that whole genome sequences of mutants were used to confirm mutants. However, there is no description of the methods used, nor can I find that those data are made available. Please add a description of the methods used for whole genome sequencing and confirming the presence of mutant alleles.

The methods for whole genome sequencing have been added in lines 551-557. The exact indel sequences for each mutant are shown in Figure 4 – supplement 3, with the guide sequence highlighted in pink. We show the sequences for both valkea and yellow-e. The raw whole genome sequences of the CRISPR mutants were deposited in SRA with the other raw data, as detailed in the data availability statement.

I am also interested in what methods were used to test for off-target effects. It is particularly important to examine for potential off-target edits to other yellow genes.

We used the whole genome data of the CRISPR mutants to look for off-target effects in the remaining yellow genes (c, d2, f, g2, g2, h and yellow). No indels or mutations were found in these gene sequences and we have added this to the manuscript (lines 233-236).

Ln 220. Only one female survived to adulthood, and this had a mosaic phenotype. "This individual had one yellow forewing, similar to the male mutants, with the rest of the body and wings being wildtype (Figure 4 —figure supplement 1)." It is not at all clear that this female has one mutant wing. Both wings appear much more yellow than a white wildtype. I need some further phenotypic evidence (spectrophotometer readings or pigment analyses) as the phenotypic variation is not evident in the images provided. It would be ideal to see that the colors in mosaic mutant phenotypic regions are significantly different from wildtype (this can be done using spec readings from multiple wildtype wings and mutant wings).

We have added more explanation about the female colour in lines 226-232. Female colour does not correlate with the male colour genotypes, and the forewings of wildtype females are a pale-yellow colour. This mutant female had one wildtype forewing and one which was much more yellow/orange than expected. This is made clearer in figure 4 – supplement 1 and also quantified in the spectral measurements, which show that the mutant forewing is closer in colour to the hindwings than to the other forewing (Figure 4 – supplement 4). We also added a photo of a wildtype female to Figure 4 – supplement 1. This shows the pale-yellow colour of the female forewings. Hindwing colour in females varies continuously from yellow/orange to red so although the hindwing colour of the mutant looks different to the wildtype, this is not unusual.

Second, there needs to be sequence verification of the mutations included in the manuscript, as previously mentioned.

The sequences for the female are also included in Figure 4 – supplement 3. We have highlighted which is the female in the figure legend.

Figure 4 —figure supplement 2 shows images UV. However, there are no methods provided for how these UV data were collected. Without some details of the imaging setup, I am unable to discern that images reflect differences in UV reflection, or may be due to variations in the imaging procedure. If possible, spectra analyses of the wings are an easy and cost-effective approach to quickly confirming changes in UV brightness on lepidoptera wings.

All photographs were taken under standard lighting conditions and images were standardised using colour standards. Detailed methods regarding the camera and filters used have been added at lines 576-582. We took spectral measurements of a set of wildtype males and females (including WW, wy and yy genotypes), and also the 5 CRISPR mutants. These plots are found in figure 4 – supplement 4 and methods in lines 583-589. The reflectance spectra show that the mutant hindwings have lost UV reflectance compared to wildtype white wings.

There is also no background information given for the wildtype UV. Lines 212-213 suggest the UV is a result of scale structures. What is the reference or evidence for this? Variation in UV reflection is known to be influenced by pigment composition in Pieris butterflies, not necessarily scale structures. To make assertions of UV being associated with scale structures, I would be interested in seeing the characterization of the putative UV related scale structures in wildtypes and mutants. This type of scale characterization (e.g. SEM and/TEM of wing scales in wildtype and mutants) is routinely included with other functional genomic studies of similar wing colorations (for examples see Ficarrotta et al. 2022, Livraghi et al. 2022, Concha et al. 2019, Matsuoka and Monteiro 2018). At a minimum, a detailed description/characterization of the wildtype UV should be given to the readers. Along these lines, I am curious to know if the UV may be iridescent. If so, some descriptive info on the iridescence would be needed (i.e. angle of incidence). Also, if iridescent, the differences in UV between wildtype and mutants should be examined further to determine if the image differences between wt and mutants are due to changes in the angle of incidence.

We are definitely interested to look more into scale structure differences and the development of these. However, at the moment little is known about how UV reflectance is produced in this species and so we are carrying out a more thorough study of this to be included in a separate manuscript. Hence in this manuscript, we make a suggestion that there could be scale structure changes, but make no firm conclusions about this and do not rule out the possibility that only pigments are affected. With only 5 mutants it would be difficult to make any statistical inferences regarding any changes. Preliminary measurements show that UV is not iridescent but again this will be part of a further analysis which we will think will benefit from being separate from this manuscript. However, the addition of the reflectance spectra now shows more clearly that the mutant hindwings resemble more closely the yellow hindwings rather than the white wings in the UV wavelengths.

Ln 335-336. I am unclear what evidence supports yellow-e having a forewing-specific effect.

Now line 350 – we state the role of yellow-e in forewing colour as a hypothesis. We have added further explanation at this point that this suggestion comes from the change in female forewing colour, since we do not expect knockouts of valkea to have any effect on female colour. Spectral measurements show that forewing colour is similar in WW, wy and yy Finnish samples, so we don’t expect the presence/absence of valkea to control forewing colour.

Ln 337-338 The authors state that yellow-e was "likely also knocked out…". I think this is misleading, as lines 218-219 states "All samples also showed evidence of editing at the corresponding yellow-e exons, which mainly involved insertions". Based on this it seems more than "likely", and actually confirmed yellow-e coding was disrupted in ALL samples.

This has been edited at the start of the discussion and also in the abstract.

Reviewer #3 (Recommendations for the authors):The revised version of the manuscript successfully addresses most of my previous concerns.Results from CrispR/cas9 experiments targeting the valkea gene were added to the manuscript in order to validate the role of this gene in the developmental switch from the yellow to the white morph. Such CrispR/cas9 experiments are challenging and obtaining high number of mutant adults is usually difficult in Lepidoptera.Here a few male mutants and one female mutant were successfully obtained. Nevertheless, the lack of specificity of the CrispR guides resulted in modifications in both valkea and yellow-e genes in the few mutant individuals that reached the adult stage, therefore preventing the full characterisation of the respective functional implications of these two genes in the development of hind and forewing colour patterns in males and females.From what I understood, the main argument for ruling out yellow-e as causing the white/yellow switch in male hindwings is the phenotype observed in a single mutant female showing in panel E of the supplementary figure 4. The sentences line 221-224 are not entirely convincing to me. The phenotype of the mutant female is used to point at the putative role of yellow-e on forewing colour in female. Does it lead to hypothesize a role of yellow-e on forewing colour in both sexes? And thus to a role of valkea in male hindwing colour? This indirect argument should be made clearer, and further discussion on the respective roles of these two genes in hind and forewing coloration is needed.

We have added some further explanation to the paragraph around lines 221-232 and hope this clarifies the hypothesis that yellow-e could affect forewing colour. We also made clearer in the discussion that knockouts of only yellow-e would be needed to confirm this hypothesis.

“As all genotypes have similar forewing colour in the wildtypes, we do not expect valkea to affect the forewing and thus the change in forewing colour could be attributed to a yellow-e mutation. Only one female survived to adulthood, and this had a mosaic phenotype. Female colour does not correlate with the male colour genotypes, and the forewings of females are a pale-yellow colour. This individual with a mosaic phenotype had one mutant forewing which was much more yellow/orange than the wildtype. The rest of the wings and body resembled a wildtype female (Figure 4 —figure supplement 1/Figure 4 —figure supplement 4). Reflectance spectra show that the mutant left forewing is closer in colour to the yellow/orange on the hindwings, than to the colour of the opposite forewing (Figure 4 —figure supplement 4). Since a valkea knockout is not expected to affect female phenotypes as they always have orange/red hindwings, this could be further evidence for the effect of yellow-e on forewing colour.”

In the supplementary figure 4 (referred to as Figure 4 —figure supplement 1): the panel E shows the phenotype of the mutant female but picture of the wild-type female would be useful to fully evaluate the impact of the CrispR treatment on phenotypic variation.

A wildtype female has been added to this figure, and spectral data for the mosaic and wildtype females can be seen in figure 4 – supplement 4.

Line 213: This is quite interesting, did you observe differences in scale structure between wild-type yellow and white scales, and in the wild-type yellow vs. mutant yellow scales? Such observations on the respective role of pigments and scale structure in the reflected colours are also relevant to understand the developmental bases of wing colour variations.

Please see the reply to reviewer 2’s comment above regarding the UV reflectance and scale structure.